# DIVERSITY ACTOR-CRITIC: SAMPLE-AWARE ENTROPY REGULARIZATION FOR SAMPLE-EFFICIENT EXPLORATION

## ABSTRACT

Policy entropy regularization is commonly used for better exploration in deep reinforcement learning (RL). However, policy entropy regularization is sample-inefficient in off-policy learning since it does not take the distribution of previous samples stored in the replay buffer into account. In order to take advantage of the previous sample distribution from the replay buffer for sample-efficient exploration, we propose sample-aware entropy regularization which maximizes the entropy of weighted sum of the policy action distribution and the sample action distribution from the replay buffer. We formulate the problem of sample-aware entropy regularized policy iteration, prove its convergence, and provide a practical algorithm named diversity actor-critic (DAC) which is a generalization of soft actor-critic (SAC). Numerical results show that DAC significantly outperforms SAC baselines and other state-of-the-art RL algorithms.

## 1 INTRODUCTION

Reinforcement learning (RL) aims to maximize the expectation of the discounted reward sum under Markov decision process (MDP) environments (Sutton & Barto, 1998). When the given task is complex, i.e. the environment has high action-dimensions or sparse rewards, it is important to well explore state-action pairs for high performance (Agre & Rosenschein, 1996). For better exploration, recent RL considers various methods: maximizing the policy entropy to take actions more uniformly (Ziebart et al., 2008; Fox et al., 2015; Haarnoja et al., 2017), maximizing diversity gain that yields intrinsic rewards to explore rare states by counting the number of visiting states (Strehl & Littman, 2008; Lopes et al., 2012), maximizing information gain (Houthooft et al., 2016; Hong et al., 2018), maximizing model prediction error (Achiam & Sastry, 2017; Pathak et al., 2017), and so on. In particular, based on policy iteration for soft Q-learning, (Haarnoja et al., 2018a) considered an off-policy actor-critic framework for maximum entropy RL and proposed the soft actor-critic (SAC) algorithm, which has competitive performance for challenging continuous control tasks.

In this paper, we reconsider the problem of policy entropy regularization in off-policy learning and propose a generalized approach to policy entropy regularization. In off-policy learning, we store and reuse old samples to update the current policy (Mnih et al., 2015), and it is preferable that the old sample distribution in the replay buffer is uniformly distributed for better performance. However, the simple policy entropy regularization tries to maximize the entropy of the current policy irrespective of the distribution of previous samples. Since the uniform distribution has maximum entropy, the current policy will choose previously less-sampled actions and more-sampled actions with the same probability and hence the simple policy entropy regularization is sample-unaware and sample-inefficient. In order to overcome this drawback, we propose sample-aware entropy regularization, which tries to maximize the weighted sum of the current policy action distribution and the sample action distribution from the replay buffer. We will show that the proposed sample-aware entropy regularization reduces to maximizing the sum of the policy entropy and the $\alpha$-skewed Jensen-Shannon divergence (Nielsen, 2019) between the policy distribution and the buffer sample action distribution, and hence it generalizes SAC. We will also show that properly exploiting the sample action distribution in addition to the policy entropy over learning phases will yield far better performance.

## 2 RELATED WORKS

**Entropy regularization**: Entropy regularization maximizes the sum of the expected return and the policy action entropy. It encourages the agent to visit the action space uniformly for each given state, and the regularized policy is robust to modeling error (Ziebart, 2010). Entropy regularization is considered in various domains for better optimization: inverse reinforcement learning (Ziebart et al., 2008), stochastic optimal control problems (Todorov, 2008; Toussaint, 2009; Rawlik et al., 2013), and off-policy reinforcement learning (Fox et al., 2015; Haarnoja et al., 2017). (Lee et al., 2019) shows that Tsallis entropy regularization that generalizes usual Shannon-entropy regularization is helpful. (Nachum et al., 2017a) shows that there exists a connection between value-based and policy-based RL under entropy regularization. (O'Donoghue et al., 2016) proposed an algorithm combining them, and it is proven that they are equivalent (Schulman et al., 2017a). The entropy of state mixture distribution is better for pure exploration than a simple random policy (Hazan et al., 2019).

**Diversity gain**: Diversity gain is used to provide a guidance for exploration to the agent. To achieve diversity gain, many intrinsically-motivated approaches and intrinsic reward design methods have been considered, e.g., intrinsic reward based on curiosity (Chentanez et al., 2005; Baldassarre & Mirolli, 2013), model prediction error (Achiam & Sastry, 2017; Pathak et al., 2017; Burda et al., 2018), divergence/information gain (Houthooft et al., 2016; Hong et al., 2018), counting (Strehl & Littman, 2008; Lopes et al., 2012; Tang et al., 2017; Martin et al., 2017), and unification of them (Bellemare et al., 2016). For self-imitation learning, (Gangwani et al., 2018) considered the Stein-variational gradient decent with the Jensen-Shannon kennel.

**Off-policy learning**: Off-policy learning can reuse any samples generated from behaviour policies for the policy update (Sutton & Barto, 1998; Degris et al., 2012), so it is sample-efficient as compared to on-policy learning. In order to reuse old samples, a replay buffer that stores trajectories generated by previous policies is used for Q-learning (Mnih et al., 2015; Lillicrap et al., 2015; Fujimoto et al., 2018; Haarnoja et al., 2018a). To enhance both stability and sample-efficiency, several methods are considered, e.g., combining on-policy and off-policy (Wang et al., 2016; Gu et al., 2016; 2017), and generalization from on-policy to off-policy (Nachum et al., 2017b; Han & Sung, 2019).

In order to guarantee the convergence of Q-learning, there is a key assumption: *Each state-action pair must be visited infinitely often* (Watkins & Dayan, 1992). If the policy does not visit diverse state-action pairs many times, it converges to local optima. Therefore, exploration for visiting different state-action pairs is important for RL, and the original policy entropy regularization encourages exploration (Ahmed et al., 2019). However, we found that the simple policy entropy regularization can be sample-inefficient in off-policy RL, so we aim to propose a new entropy regularization method that significantly enhances the sample-efficiency for exploration by considering the previous sample distribution in the buffer.

## 3 BACKGROUND

In this section, we briefly introduce the basic setup and the soft actor-critic (SAC) algorithm.

### 3.1 SETUP

We assume a basic RL setup composed of an environment and an agent. The environment follows an infinite horizon Markov decision process $(\mathcal{S}, \mathcal{A}, P, \gamma, r)$, where $\mathcal{S}$ is the state space, $\mathcal{A}$ is the action space, $P$ is the transition probability, $\gamma$ is the discount factor, and $r : \mathcal{S} \times \mathcal{A} \to \mathbb{R}$ is the reward function. In this paper, we consider a continuous state-action space. The agent has a policy distribution $\pi : \mathcal{S} \times \mathcal{A} \to [0, \infty)$ which selects an action $a_t$ for a given state $s_t$ at each time step $t$, and the agent interacts with the environment and receives reward $r_t := r(s_t, a_t)$ from the environment. Standard RL aims to maximize the discounted return $\mathbb{E}_{s_0 \sim p_0, \tau_0 \sim \pi}[\sum_{t=0}^{\infty} \gamma^t r_t]$, where $\tau_t = (s_t, a_t, s_{t+1}, a_{t+1} \cdots)$ is an episode trajectory.

### 3.2 SOFT ACTOR-CRITIC

Soft actor-critic (SAC) (Haarnoja et al., 2018a) includes a policy entropy regularization term in the objective function for better exploration by visiting the action space uniformly for each given state.

The entropy-augmented policy objective function of SAC is given by

$$J_{SAC}(\pi) = \mathbb{E}_{\tau_0 \sim \pi} \left[ \sum_{t=0}^{\infty} \gamma^t (r_t + \beta \mathcal{H}(\pi(\cdot|s_t))) \right], \tag{1}$$

where $\mathcal{H}$ is the entropy function and $\beta \in (0, \infty)$ is the entropy coefficient. SAC is a practical off-policy actor-critic based on soft policy iteration (SPI) that alternates soft policy evaluation to estimate the true soft $Q$-function and soft policy improvement to find the optimal policy that maximizes (1). In addition, SPI theoretically guarantees convergence to the optimal policy that maximizes (1).

## 4 THE DIVERSITY ACTOR-CRITIC ALGORITHM

### 4.1 MOTIVATION OF THE SAMPLE-AWARE ENTROPY

As explained in Section 2, the policy should visit diverse samples to learn the policy without converging to the local optima. In off-policy learning, we can reuse previous samples stored in the replay buffer to learn the policy, so it is efficient to draw diverse samples while avoiding frequently selected samples before. The policy entropy maximization enhances exploration to yield better performance, but it is sample-inefficient for off-policy RL because it does not take advantage of the previous sample action distribution obtainable from the replay buffer: If we assume bounded action space, the simple policy entropy maximization will choose all actions with the equal probability without considering the previous action samples because $\max_\pi \mathcal{H}(\pi) = \min_\pi \mathcal{D}_{KL}(\pi||U)$ is achieved when $\pi = U$, where $U$ is a uniform distribution and $\mathcal{D}_{KL}$ is the Kullback-Leibler (KL) divergence.

In order to overcome the limitation of the simple policy entropy maximization, we consider maximizing *a sample-aware entropy* defined as the entropy of a mixture distribution of the policy distribution $\pi$ and the current sample action distribution $q$ in the replay buffer. Here, $q$ is defined as

$$q(\cdot|s) := \frac{\sum_{a \in \mathcal{D}} N(s,a)\delta_a(\cdot)}{\sum_{a' \in \mathcal{D}} N(s,a')}, \tag{2}$$

where $\mathcal{D}$ is the replay buffer that stores previous samples $(s_t, a_t, r_t, s_{t+1})$ at each time $t$, $\delta_a(\cdot)$ is the Dirac measure at $a \in \mathcal{A}$, and $N(s,a)$ is the number of state-action pair $(s,a)$ in $\mathcal{D}$.

Then, we define a target distribution $q_{target}^{\pi,\alpha}$ as the mixture distribution of $\pi$ and $q$, which is expressed as $q_{target}^{\pi,\alpha} := \alpha\pi + (1-\alpha)q$, where $\alpha \in [0,1]$ is the weighting factor. Note that we draw samples from policy $\pi$ and store them in the replay buffer, so the target distribution can be viewed as the updated sample action distribution in the future replay buffer. Then, maximizing the sample-aware entropy $\mathcal{H}(q_{target}^{\pi,\alpha})$ can encourage sample-efficient exploration because $\pi$ will choose actions rare in the buffer with high probability and actions stored many times in the buffer with low probability in order to make the target distribution uniform. We provide a simple example below:

Let us consider a simple 1-step MDP in which $s_0$ is the unique initial state, there exist $N_a$ actions ($\mathcal{A} = \{A_0, \cdots, A_{N_a-1}\}$), $s_1$ is the terminal state, and $r$ is a deterministic reward function. Then, there exist $N_a$ state-action pairs in total and let us assume that we already have $N_a - 1$ state-action samples in the replay buffer as $\mathbf{R} = \{(s_0, A_0, r(s_0, A_0)), \cdots, (s_0, A_{N_a-2}, r(s_0, A_{N_a-2}))\}$. In order to estimate the Q-function for all state-action pairs, the policy should sample the last action $A_{N_a-1}$ (After then, we can reuse all samples infinitely to estimate $Q$). Here, we will compare two exploration methods.

1) First, if we consider the simple entropy maximization, the policy that maximizes its entropy will choose all actions with equal probability $1/N_a$ (uniformly). Then, $N_a$ samples should be taken on average by the policy to visit the action $A_{N_a-1}$.

2) Consider the sample-aware entropy maximization. Here, the sample action distribution $q$ in the buffer becomes $q(a_0|s_0) = 1/(N_a - 1)$ for $a_0 \in \{A_0, \cdots, A_{N_a-2}\}$ and $q(A_{N_a-1}|s_0) = 0$, the target distribution becomes $q_{target}^{\pi,\alpha} = \alpha\pi + (1-\alpha)q$, and we set $\alpha = 1/N_a$. Then, the policy that maximizes the sample-aware entropy becomes $\pi(A_{N_a-1}|s_0) = 1$ to make $q_{target}^{\pi,\alpha}$ uniform because $\max_\pi \mathcal{H}(q_{target}^{\pi,\alpha}) = \min_\pi \mathcal{D}_{KL}(q_{target}^{\pi,\alpha}||U)$. In this case, we only needs one sample to visit the

action $A_{N_a-1}$. In this way, the simple entropy maximization is sample-inefficient for off-policy RL, and the proposed sample-aware entropy maximization can enhance the sample-efficiency for exploration by using the previous sample distribution and choosing a proper $\alpha$. With this motivation, we propose the sample-aware entropy regularization for off-policy RL and the corresponding $\alpha$-adaptation method.

## 4.2 SAMPLE-AWARE ENTROPY REGULARIZATION

Our approach is to maximize the return while maximizing the sample-aware entropy. Under this approach, previously many times sampled actions will be given low probabilities and previously less taken actions will be given high probabilities by the current policy $\pi$ for sample-efficient exploration as shown in Section 4.1. Hence, we set the objective function for the proposed sample-aware entropy regularization as

$$J(\pi) = \mathbb{E}_{\tau_0 \sim \pi}\left[\sum_{t=0}^{\infty} \gamma^t(r_t + \beta\mathcal{H}(q_{target}^{\pi,\alpha}(\cdot|s_t)))\right]. \tag{3}$$

Here, the sample-aware entropy $\mathcal{H}(q_{target}^{\pi,\alpha})$ for given $s_t$ can be decomposed as

$$\mathcal{H}(q_{target}^{\pi,\alpha}) = -\int_{a\in\mathcal{A}}(\alpha\pi + (1-\alpha)q)\log(\alpha\pi + (1-\alpha)q) = \alpha\mathcal{H}(\pi) + D_{JS}^\alpha(\pi||q) + (1-\alpha)\mathcal{H}(q), \tag{4}$$

where $D_{JS}^\alpha(\pi||q) := \alpha D_{KL}(\pi||q_{target}^{\pi,\alpha}) + (1-\alpha)D_{KL}(q||q_{target}^{\pi,\alpha})$ is the $\alpha$ skew-symmetric Jensen-Shannon (JS) divergence (Nielsen, 2019). Note that $D_{JS}^\alpha$ reduces to the standard JS divergence for $\alpha = \frac{1}{2}$ and to zero for $\alpha = 0$ or 1. Hence, for $\alpha = 1$, (4) reduces to the simple entropy, but for $\alpha \neq 1$, it is a generalization incorporating the distribution $q$. Thus, our objective function aims to maximize the return while simultaneously maximizing the discounted sum of policy entropy, sample entropy, and the divergence between $\pi$ and $q$. In this way, the policy will choose more diverse actions that are far from the samples stored in the replay buffer while maintaining its entropy for better exploration.

## 4.3 DIVERSE POLICY ITERATION WITH THE PROPOSED OBJECTIVE

In this section, we derive the diverse policy evaluation and diverse policy improvement to maximize the objective function with the sample-aware entropy regularization (3). Note that the sample action distribution $q$ is updated as the iteration goes on. However, it changes very slowly since the buffer size is much larger than the time steps of one iteration. Hence, for the purpose of proof we regard the action distribution $q$ as a fixed distribution in this section.

First, we define the true diverse $Q$-function $Q^\pi$ as $Q^\pi(s_t, a_t) := \frac{1}{\beta}r_t + \mathbb{E}_{\tau_{t+1}\sim\pi}\left[\sum_{l=t+1}^{\infty}\gamma^{l-t-1}\left(\frac{1}{\beta}r_l + \alpha\mathcal{H}(\pi(\cdot|s_l)) + D_{JS}^\alpha(\pi(\cdot|s_l)||q(\cdot|s_l)) + (1-\alpha)\mathcal{H}(q(\cdot|s_l))\right)\right].$

We defined the sample distribution $q$ in equation (2), but we do not want to compute actual $q$, which requires a method such as discretization and counting for continuous samples. Even if $q$ is obtained by counting, a generalization of $q$ for arbitrary state-action pairs is needed again to estimate $Q^\pi$. We circumvented this difficulty by defining the ratio $R^{\pi,\alpha}$ of $\alpha\pi$ to $q_{target}^{\pi,\alpha}$ as

$$R^{\pi,\alpha}(s_t, a_t) = \frac{\alpha\pi(a_t|s_t)}{\alpha\pi(a_t|s_t) + (1-\alpha)q(a_t|s_t)}, \tag{5}$$

and we will show later that all objective (or loss) functions for practical implementation can be represented by using the ratio only, without using the explicit $q$ in Appendix B.

Then, we can decompose $D_{JS}^\alpha(\pi(\cdot|s_l)||q(\cdot|s_l))$ as

$$D_{JS}^\alpha(\pi||q) = \alpha\mathbb{E}_{a_l\sim\pi(\cdot|s_l)}[\log R^{\pi,\alpha}(s_l, a_l)] + (1-\alpha)\mathbb{E}_{a_l\sim q(\cdot|s_l)}[\log(1 - R^{\pi,\alpha}(s_l, a_l))] + H(\alpha), \tag{6}$$

where $H(\alpha) = -\alpha\log\alpha - (1-\alpha)\log(1-\alpha)$ is the binary entropy function.

The modified Bellman backup operator for $Q^\pi$ estimation is given by

$$\mathcal{T}^\pi Q(s_t, a_t) := \frac{1}{\beta}r_t + \gamma\mathbb{E}_{s_{t+1}\sim P}[V(s_{t+1})], \tag{7}$$

where $V(s_t) = \mathbb{E}_{a_t \sim \pi}[Q(s_t, a_t) + \alpha \log R^{\pi, \alpha}(s_t, a_t) - \alpha \log \alpha \pi(a_t|s_t)] + (1 - \alpha)\mathbb{E}_{a_t \sim q}[\log(1 - R^{\pi, \alpha}(s_t, a_t)) - \log(1 - \alpha)q(a_t|s_t)]$ is an estimated diverse state value function, $Q : \mathcal{S} \times \mathcal{A} \to \mathbb{R}$ is an estimated diverse state-action value function.

Proof of the convergence of diverse policy evaluation that estimates $Q^\pi$ by repeating the Bellman operator (7) is provided in Appendix A. Then, the policy is updated from $\pi_{old}$ to $\pi_{new}$ as $\pi_{new} = \arg\max_\pi J_{\pi_{old}}(\pi)$, where $J_{\pi_{old}}(\pi)$ is the objective of $\pi$ estimated under $Q^{\pi_{old}}$ defined as[1]

$$J_{\pi_{old}}(\pi(\cdot|s_t)) := \beta\{\mathbb{E}_{a_t \sim \pi}[Q^{\pi_{old}}(s_t, a_t) + \alpha \log R^{\pi, \alpha}(s_t, a_t) - \alpha \log \alpha\pi(a_t|s_t)]$$
$$+ (1 - \alpha)\mathbb{E}_{a_t \sim q}[\log(1 - R^{\pi, \alpha}(s_t, a_t)) - \log(1 - \alpha)q(a_t|s_t)]\}. \quad (8)$$

The monotone improvement of this step is proved in Appendix A. Now, we can find the optimal policy that maximizes $J(\pi)(= J_\pi(\pi))$ by the following theorem:

**Theorem 1 (Diverse Policy Iteration)** *By repeating iteration of the diverse policy evaluation and the diverse policy improvement, any initial policy converges to the optimal policy $\pi^*$ s.t. $Q^{\pi^*}(s_t, a_t) \geq Q^{\pi'}(s_t, a_t), \forall \pi' \in \Pi, \forall (s_t, a_t) \in \mathcal{S} \times \mathcal{A}$. Also, such $\pi^*$ achieves maximum $J$, i.e., $J_{\pi^*}(\pi^*) \geq J_\pi(\pi)$ for any $\pi \in \Pi$.*

*Proof.* See Appendix A.1.

Note that $J_{\pi_{old}}(\pi)$ for diverse policy iteration above requires the ratio function $R^{\pi, \alpha}$ of the current policy $\pi$, but we can only estimate $R^{\pi_{old}, \alpha}$ for the previous policy $\pi_{old}$ in practice. Thus, we circumvent this difficulty by defining a practical objective function $\tilde{J}_{\pi_{old}}(\pi)$ given by

$$\tilde{J}_{\pi_{old}}(\pi(\cdot|s_t)) := \beta\mathbb{E}_{a_t \sim \pi}[Q^{\pi_{old}}(s_t, a_t) + \alpha \log R^{\pi_{old}, \alpha}(s_t, a_t) - \alpha \log \pi(a_t|s_t)], \quad (9)$$

Regarding the practically computable objective function $\tilde{J}_{\pi_{old}}(\pi)$, we have the following result:

**Theorem 2** *Suppose that the policy is parameterized with parameter $\theta$. For parameterized policy $\pi_\theta$, two objective functions $J_{\pi_{\theta_{old}}}(\pi_\theta(\cdot|s_t))$ and $\tilde{J}_{\pi_{\theta_{old}}}(\pi_\theta(\cdot|s_t))$ have the same gradient direction for $\theta$ at $\theta = \theta_{old}$ for all $s_t \in \mathcal{S}$.*

*Proof.* See Appendix A.2.

By Theorem 2, we can replace the objective function $J_{\pi_{old}}(\pi)$ of policy improvement with the practically computable objective function $\tilde{J}_{\pi_{old}}(\pi)$ for parameterized policy without loss of optimality.

## 4.4 DIVERSITY ACTOR CRITIC IMPLEMENTATION

We first define $R^\alpha$ as an estimate for the ratio function $R^{\pi_{old}, \alpha}$. For implementation, we parameterize $\pi$, $R^\alpha$, $Q$, and $V$ by neural network parameters $\theta$, $\eta$, $\phi$, and $\psi$, respectively. Then, we setup the practical objective (or loss) functions $\hat{J}_\pi(\theta)$, $\hat{J}_{R^\alpha}(\eta)$, $\hat{L}_Q(\phi)$, and $\hat{L}_V(\psi)$ for the parameter update. Detailed DAC implementation based on Section 4 is provided in Appendix B. The proposed DAC algorithm is summarized in Appendix C. Note that DAC becomes SAC when $\alpha = 1$, and becomes standard off-policy RL without entropy regularization when $\alpha = 0$.

## 5 $\alpha$-ADAPTATION

In the proposed sample-aware entropy regularization, the weighting factor $\alpha$ plays an important role in controlling the ratio between the policy distribution $\pi$ and the sample action distribution $q$. However, it is difficult to estimate optimal $\alpha$ directly. Hence, we further propose an adaptation method for $\alpha$ based on max-min principle widely considered in game theory, robust learning, and decision making problems (Chinchuluun et al., 2008). Since we do not know optimal $\alpha$, an alternative formulation is that we maximize the return while maximizing the worst-case sample-aware entropy, i.e., $\min_\alpha \mathcal{H}(q_{target}^{\pi, \alpha})$. Then, the max-min approach can be formulated as follows:

$$\max_\pi \mathbb{E}_{\tau_0 \sim \pi}\left[\sum_t \gamma^t(r_t + \beta \min_\alpha[\mathcal{H}(q_{target}^{\pi, \alpha}) - \alpha c])\right] \quad (10)$$

---

[1]Note that if we replace $\pi_{old}$ with $\pi$ and view every state $s_t$ as an initial state, then (8) reduces to $J(\pi)$.

where $c$ is a control hyperparameter for $\alpha$ adaptation. We learn $\alpha$ to minimize $\mathcal{H}(q_{target}^{\pi,\alpha}) - \alpha c$, so the role of $c$ is to maintain the target entropy at a certain level to explore the state-action well. Detailed implementation for $\alpha$-adaptation is given in Appendix B.1.

## 6 EXPERIMENTS

In this section, we evaluate the proposed DAC algorithm on various continuous-action control tasks and provide ablation study. In order to see the superiority of the sample-aware entropy regularization, we here focus on comparison with two SAC baselines: SAC and SAC-Div. SAC-Div is SAC combined with the method in (Hong et al., 2018) that diversifies policies from buffer distribution by simply maximizing $J(\pi) + \alpha_d D(\pi||q)$ for $J(\pi)$ in (1) and some divergence $D$. Note that the key difference between SAC-Div and DAC is that SAC-Div simply adds the single divergence term to the policy objective function $J(\pi)$, whereas DAC considers the discounted sum of target entropy terms as seen in (3). For SAC-Div, we consider KL divergence (MSE if the policy is Gaussian) and adaptive scale $\alpha_d$ with $\delta_d = 0.2$ for the divergence term as suggested in (Hong et al., 2018). In order to rule out the influence of factors other than exploration, we use the common simulation setup for DAC and SAC baselines except for the parts about entropy or divergence.

In addition, we provide comparison of DAC to random network distillation (RND) (Burda et al., 2018) and MaxEnt (Hazan et al., 2019), which are the recent exploration methods based on finding rare states in Appendix F.2, and to other recent RL algorithms in Appendix F.3. The result shows that DAC yields the best performance for all considered tasks as compared to recent RL algorithms. We also provide the source code of DAC implementation that requires Python Tensorflow. Detailed simulation setup for experiments is summarized in Appendix E.

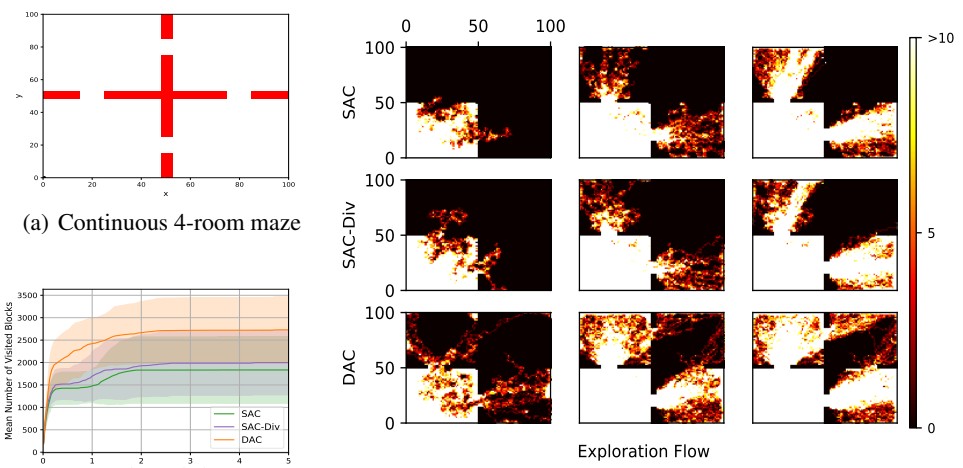

(a) Continuous 4-room maze

(b) Number of state visitation

(c) State visit histogram at 5k (left) 50k (middle) 500k (right) steps

Figure 1: Pure exploration task: Continuous 4-room maze

### 6.1 PURE EXPLORATION COMPARISON

In order to see the exploration performance of DAC ($\alpha = 0.5$) as compared to the SAC baselines, we compare state visitation on a $100 \times 100$ continuous 4-room maze task. The maze environment is made by modifying a continuous grid map available at https://github.com/huyaoyu/GridMap, and it is shown in Fig. 1(a). State is $(x, y)$ position in the maze, action is $(dx, dy)$ bounded by $[-1, 1]$, and the agent location after the action becomes $(x + dx, y + dy)$. The agent starts from the left lower corner $(0.5, 0.5)$ and explores the maze without any reward, and Fig. 1(b) shows the mean number of new state visitations over 30 seeds, where the number of state visitation is obtained for each integer interval. As seen in Fig. 1(b), DAC visited much more states than SAC/SAC-Div, which means that the exploration performance of DAC is superior to that of the SAC baselines. In addition, Fig. 1(c) shows the corresponding state visit histogram of all seeds. Here, as the color of the state becomes brighter, the state is visited more times. Note that SAC/SAC-Div rarely visit the right upper room even at 500k time steps for all seeds, but DAC starts visiting the

right upper room at 5k time steps and frequently visit the right upper room at 500k time steps. Thus, Fig. 1(c) clearly shows that DAC has better sample-efficiency for exploration than SAC/SAC-Div.

## 6.2 PERFORMANCE COMPARISON WITH THE SAC BASELINES

The final goal of RL is to achieve high scores for given tasks. For this, exploration techniques are needed to ensure that the policy does not converge to local optima, as explained in Section 2. We first showed the improvement of the exploration performance in a pure exploration task (continuous 4-room maze), and experiments in this section will show that DAC has better return performance than SAC baselines on several sparse-rewarded tasks. Note that having high scores on the sparse-reward tasks means that the policy can get rewards well without falling into local optima, which implies that the agent successfully explores more state-action pairs that have positive (or diverse) rewards. Therefore, the performance comparison on sparse tasks fits well to the motivation and also note that sparse-rewarded tasks has been widely used as a verification method of exploration in many previous exploration studies (Hong et al., 2018; Mazoure et al., 2019; Burda et al., 2018).

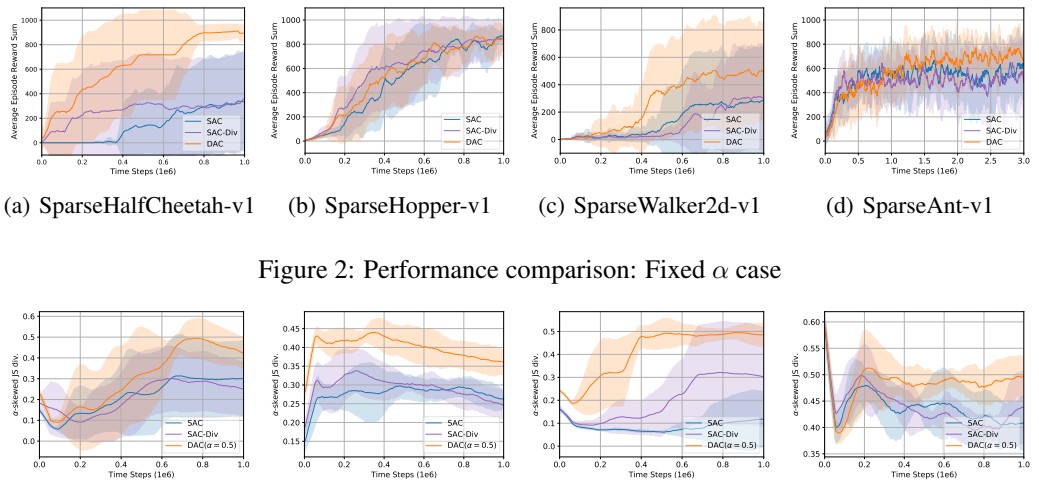

| (a) SparseHalfCheetah-v1 | (b) SparseHopper-v1 | (c) SparseWalker2d-v1 | (d) SparseAnt-v1 |

Figure 2: Performance comparison: Fixed $\alpha$ case

| (a) SparseHalfCheetah-v1 | (b) SparseHopper-v1 | (c) SparseWalker-v1 | (d) SparseAnt-v1 |

Figure 3: $\alpha$-skewed JS divergence for DAC and SAC/SAC-Div

**Fixed $\alpha$ case**: In order to see the advantage of the sample-aware entropy regularization for rewarded tasks, we compare the performance of DAC with $\alpha = 0.5$ and the SAC baselines on simple MDP tasks: SparseMujoco tasks. SparseMujoco is a sparse version of Mujoco and the reward is 1 if the agent exceeds the x-axis threshold, otherwise 0 (Hong et al., 2018; Mazoure et al., 2019).

The performance results averaged over 10 random seeds are shown in Fig. 2. As seen in Fig. 2, DAC has significant performance gain for most tasks as compared to SAC. On the other hand, SAC-Div also enhances the convergence speed compared to SAC for some tasks, but it fails to enhance the final performance. Fig. 3 shows the $\alpha$-skewed JS divergence curve ($\alpha = 0.5$) of DAC and SAC/SAC-Div for sparse Mujoco tasks and we provide Fig. F.1 in Appendix F.1 that shows the corresponding mean number of discretized state visitation curve on sparse Mujoco tasks. For SAC/SAC-Div, the ratio function $R$ is estimated separately from (B.2) in Appendix B and the divergence is computed from $R$. The performance table for all tasks is given by Table F.1 in Appendix F.1. As seen in Fig. 3, the divergence of DAC is much higher than that of SAC/SAC-Div throughout the learning time. It means that the policy of DAC choose more diverse actions from the distribution far away from the sample action distribution $q$, then DAC visits more diverse states than the SAC baselines as seen in Fig. F.1. Thus, DAC encourages better exploration and it yields better performance. Thus, we can conclude that the proposed sample-aware entropy regularization is superior to the simple policy entropy regularization of SAC and single divergence regularization of SAC-Div in terms of exploration and the convergence.

**Adaptive $\alpha$ case**: Now, we compare the performance of DAC with $\alpha = 0.5$, $0.8$, $\alpha$-adaptation, and the SAC baselines to see the need of $\alpha$-adaptation. To maintain controllability and prevent saturation

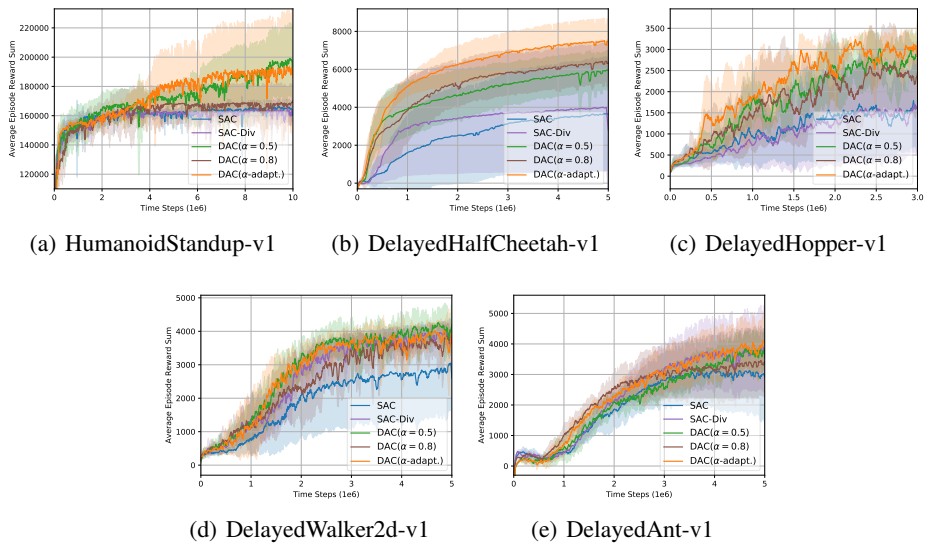

(a) HumanoidStandup-v1          (b) DelayedHalfCheetah-v1          (c) DelayedHopper-v1

(d) DelayedWalker2d-v1          (e) DelayedAnt-v1

Figure 4: Performance comparison: Adaptive $\alpha$ case

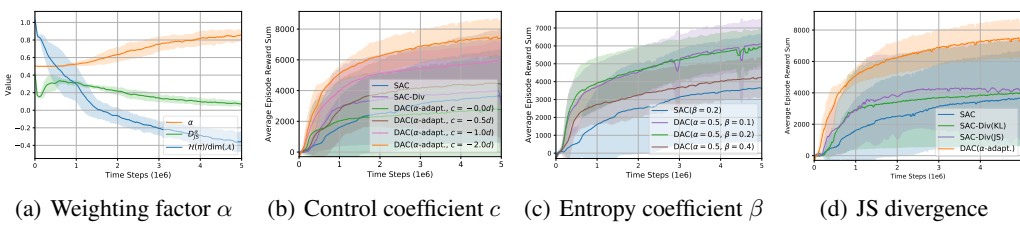

(a) Weighting factor $\alpha$     (b) Control coefficient $c$     (c) Entropy coefficient $\beta$     (d) JS divergence

Figure 5: Averaged learning curve for ablation study

of $R_\eta^\alpha$, we used regularization for $\alpha$ learning and restricted the range of $\alpha$ as $0.5 \le \alpha \le 0.99$ for $\alpha$ adaptation so that a certain level of entropy regularization is enforced. Here, we consider more complicated tasks: HumanoidStandup and delayed Mujoco tasks (DelayedHalfCheetah, Delayed-Hopper, DelayedWalker2d, and DelayedAnt). HumanoidStandup is one of high-action dimensional Mujoco tasks. Delayed Mujoco tasks suggested by (Zheng et al., 2018; Guo et al., 2018) have the same state-action spaces with original Mujoco tasks but reward is sparsified. That is, rewards for $D$ time steps are accumulated and the accumulated sum is delivered to the agent once every $D$ time steps, so the agent receives no reward during the accumulation time. The performance results averaged over 5 random seeds are shown in Fig. 4. The result of the max average return of these Mujoco tasks for DAC and SAC/SAC-Div is provided in Table F.2 in Appendix F.1. As seen in Fig. 4, all versions of DAC outperform SAC. Here, SAC-Div also outperforms SAC for several tasks, but the performance gain by DAC is much higher. In addition, it is seen that the best $\alpha$ depends on the tasks in the fixed $\alpha$ case. For example, $\alpha = 0.8$ is the best for DelayedHalfCheetah, but $\alpha = 0.5$ is the best for DelayedAnt. Thus, we need to adapt $\alpha$ for each task. Finally, DAC with $\alpha$-adaptation has the top-level performance for most tasks and the best performance for HumanoidStandup and DelayedHopper tasks. Further consideration for $\alpha$ is provided in Section 6.3.

## 6.3 ABLATION STUDY

In this section, we provide ablation study for important parameters in the sample-aware entropy regularization on the DelayedHalfCheetah task. Ablation studies on the other DelayedMucoco tasks are provided in Appendix G.

**Weighting factor $\alpha$:** As seen in Section 6.2, $\alpha$-adaptation is necessary because one particular value of $\alpha$ is not best for all environments. Although the proposed $\alpha$-adaptation in Section 5 is sub-optimal, it shows good performance across all the considered tasks. Thus, we study more on the proposed $\alpha$-adaptation and the sensible behavior of sample-awareness in entropy regularization.

Fig. 5(a) shows the averaged learning curve of $\alpha$, $\alpha$-skewed JS divergence $D_{JS}(\pi||q)$ and the policy entropy $\mathcal{H}(\pi)$ for DAC with the proposed $\alpha$-adaptation method on DelayedHalfCheetah. Here, we fix the control coefficient $c$ as $-2.0\dim(\mathcal{A})$. As seen in (3), the return, the policy entropy and the JS divergence are intertwined in the cost function, so their learning curves are also intertwined over time steps. Here, the learned policy entropy term decreases and the learned $\alpha$ increases to one as time step goes on. Then, the initially nonzero JS divergence term $D_{JS}(\pi||q)$ diminishes to zero, which means that the sample action distribution is exploited for roughly initial 2.5M time steps, and then DAC operates like SAC. This adaptive exploitation of the sample-aware entropy leads to better overall performance across time steps as seen in Fig. 4, so DAC with $\alpha$-adaptation seems to properly exploit both the policy entropy and the sample action distribution depending on the learning stage.

**Control coefficient** $c$: In the proposed $\alpha$-adaptation (10), the control coefficient $c$ affects the learning behavior of $\alpha$. Since $\mathcal{H}(\pi)$ and $D_{JS}^{\alpha}$ are proportional to the action dimension, we tried a few values such as $0$, $-0.5d$, $-1.0d$ and $-2.0d$ where $d = \dim(\mathcal{A})$. Fig. 5(b) shows the corresponding performance of DAC with $\alpha$-adaptation on DelayedHalfCheetah. As seen in Fig. 5(b), the performance depends on the change of $c$ as expected, and $c = -2.0 \cdot \dim(\mathcal{A})$ performs well. We observed that $-2.0d$ performed well for all considered tasks, thus we set $c = -2.0d$ in (B.8).

**Entropy coefficient** $\beta$: As mentioned in (Haarnoja et al., 2018a), the performance of SAC depends on $\beta$. It is expected that the performance of DAC depends on $\beta$ too. Fig 5(c) shows the performance of DAC with fixed $\alpha = 0.5$ for three different values of $\beta$: $\beta = 0.1$, $0.2$ and $0.4$ on Delayed-HalfCheetah. It is seen that the performance of DAC indeed depends on $\beta$. Although there exists performance difference for DAC depending on $\beta$, the performance of DAC is much better than SAC for a wide range of $\beta$. One thing to note is that the coefficient of pure policy entropy regularization term for DAC is $\alpha\beta$, as seen in (3). Thus, DAC with $\alpha = 0.5$ and $\beta = 0.4$ has the same amount of pure policy entropy regularization as SAC with $\beta = 0.2$. However, DAC with $\alpha = 0.5$ and $\beta = 0.4$ has much higher performance than SAC with $\beta = 0.2$, as seen in Fig. 5(c). So, we can see that the performance improvement of DAC comes from joint use of policy entropy $\mathcal{H}(\pi)$ and the sample action distribution from the replay buffer via $D_{JS}^{\alpha}(\pi||q)$.

**The effect of JS divergence**: In order to see the effect of the JS divergence on the performance, we also provide an additional ablation study that we consider a single JS divergence for SAC-Div by using the ratio function in Section 4.3. 5(d) shows the performance comparison of SAC, SAC-Div(KL), SAC-Div(JS), and DAC. For SAC-Div(JS), we used $\delta_d = 0.5$ for adaptive scaling in (Hong et al., 2018). As a result, there was no significant difference in performance between SAC-Div with JS divergence and SAC-Div with KL divergence. On the other hand, the DAC still shows a greater performance increase than both SAC-Div(KL) and SAC-Div(JS), and this means that the DAC has more advantages than simply using JS divergence.

## 7 CONCLUSION AND FUTURE WORKS

In this paper, we have proposed a sample-aware entropy framework for off-policy RL to overcome the limitation of simple policy entropy for sample-efficient exploration. With the sample-aware entropy regularization, we can achieve diversity gain by exploiting sample history in the replay buffer in addition to policy entropy. For practical implementation of sample-aware entropy regularized policy optimization, we have proposed the DAC algorithm with convergence proof. We have also provided an adaptation method for DAC to control the ratio of the sample action distribution to the policy action entropy. DAC is an actor-critic algorithm for sample-aware regularized policy optimization and generalizes SAC. Numerical results show that DAC significantly outperforms SAC baselines in Maze exploration and various Mujoco tasks.

For further study, we consider a generalization of our method in order to deal with the entropy of the state-action distribution. Currently, many recent papers only consider one of the entropy of state distribution $d^{\pi}(s)$ or that of action distribution $\pi(a|s)$ only since they have much different properties (e.g. the state-based entropy is non-convex on $\pi$ and the action-based entropy is convex on $\pi$). However, both entropies can be handled simultaneously as one fused entropy that deals with the entropy of the state-action distribution, factorized as $\log d^{\pi}(s, a) = \log d^{\pi}(s) + \log \pi(a|s)$. Then, the generalization of our method for the fused entropy may be able to further enhance the exploration performance by considering the exploration on the entire state-action space.

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

# A   PROOFS

## A.1   PROOF OF THEOREM 1

For a fixed policy $\pi$, $Q^\pi$ can be estimated by repeating the Bellman backup operator by Lemma 1. Lemma 1 is based on usual policy evaluation but has a new ingredient of the ratio condition in the sample-aware case.

**Lemma 1 (Diverse Policy Evaluation)** *Define a sequence of diverse Q-functions as $Q_{k+1} = \mathcal{T}^\pi Q_k$, $k \geq 0$, where $\pi$ is a fixed policy and $Q_0$ is a real-valued initial Q. Assume that the action space is bounded, and $R^{\pi,\alpha}(s_t, a_t) \in (0, 1)$ for all $(s_t, a_t) \in \mathcal{S} \times \mathcal{A}$. Then, the sequence $\{Q_k\}$ converges to the true diverse state-action value $Q^\pi$.*

*Proof.* Let $r_{\pi,t} := \frac{1}{\beta} r_t + \gamma \mathbb{E}_{s_{t+1} \sim P}[\mathbb{E}_{a_{t+1} \sim \pi}[\alpha \log R^{\pi,\alpha}(s_{t+1}, a_{t+1}) - \alpha \log \alpha \pi(a_{t+1}|s_{t+1})] + (1 - \alpha) \mathbb{E}_{a_{t+1} \sim q}[\log(1 - R^{\pi,\alpha}(s_{t+1}, a_{t+1})) - \log(1 - \alpha) q(a_{t+1}|s_{t+1})]]$. Then, we can formulate the standard Bellman equation form for the true $Q^\pi$ as

$$\mathcal{T}^\pi Q(s_t, a_t) = r_{\pi,t} + \gamma \mathbb{E}_{s+1 \sim P, \, a_{t+1} \sim \pi}[Q(s_{t+1}, a_{t+1})] \tag{A.1}$$

Under the assumption of a bounded action space and $R^{\pi,\alpha} \in (0, 1)$, the reward $r_{\pi,t}$ is bounded and the convergence is guaranteed as the usual policy evaluation (Sutton & Barto, 1998; Haarnoja et al., 2018a).

Now, we prove diverse policy improvement in Lemma 2 and diverse policy iteration in Theorem 1 by using $J_{\pi_{old}}(\pi)$ in a similar way to usual RL or SAC.

**Lemma 2 (Diverse Policy Improvement)** *Let $\pi_{new}$ be the updated policy obtained by solving $\pi_{new} = \arg\max\limits_{\pi \in \Pi} J_{\pi_{old}}(\pi)$. Then, $Q^{\pi_{new}}(s_t, a_t) \geq Q^{\pi_{old}}(s_t, a_t), \, \forall \, (s_t, a_t) \in \mathcal{S} \times \mathcal{A}$.*

*Proof.* We update the policy to maximize $J_{\pi_{old}}(\pi)$, so $J_{\pi_{old}}(\pi_{new}) \geq J_{\pi_{old}}(\pi_{old})$. Hence,

$$\begin{aligned}
&\mathbb{E}_{a_t \sim \pi_{new}}[Q^{\pi_{old}}(s_t, a_t) + \alpha \log R^{\pi_{new},\alpha}(s_t, a_t) - \alpha \log \alpha \pi_{new}(a_t|s_t)] \\
&\qquad + (1 - \alpha) \mathbb{E}_{a_t \sim q}[\log(1 - R^{\pi_{new},\alpha}(s_t, a_t)) - \log(1 - \alpha) q(a_t|s_t)] \\
\geq &\mathbb{E}_{a_t \sim \pi_{old}}[Q^{\pi_{old}}(s_t, a_t) + \alpha \log R^{\pi_{old},\alpha}(s_t, a_t) - \alpha \log \alpha \pi_{old}(a_t|s_t)] \\
&\qquad + (1 - \alpha) \mathbb{E}_{a_t \sim q}[\log(1 - R^{\pi_{old},\alpha}(s_t, a_t)) - \log(1 - \alpha) q(a_t|s_t)] \\
= &V^{\pi_{old}}(s_t)
\end{aligned} \tag{A.2}$$

By repeating the Bellman equation (7) and (A.2) at $Q^{\pi_{old}}$,

$$\begin{aligned}
Q^{\pi_{old}}(s_t, a_t) &= \frac{1}{\beta} r_t + \gamma \mathbb{E}_{s_{t+1} \sim P}[V^{\pi_{old}}(s_{t+1})] \\
&\leq \frac{1}{\beta} r_t + \gamma \mathbb{E}_{s_{t+1} \sim P}[\mathbb{E}_{a_{t+1} \sim \pi_{new}}[Q^{\pi_{old}}(s_{t+1}, a_{t+1}) + \alpha \log R^{\pi_{new},\alpha}(s_{t+1}, a_{t+1}) \\
&\quad - \alpha \log \alpha \pi_{new}(a_{t+1}|s_{t+1})] + (1 - \alpha) \mathbb{E}_{a_{t+1} \sim q}[\log(1 - R^{\pi_{new},\alpha}(s_{t+1}, a_{t+1})) \\
&\quad - \log(1 - \alpha) q(a_{t+1}|s_{t+1})]] \\
&\;\;\vdots \\
&\leq Q^{\pi_{new}}(s_t, a_t),
\end{aligned} \tag{A.3}$$

for each $(s_t, a_t) \in \mathcal{S} \times \mathcal{A}$.

**Theorem 1 (Diverse Policy Iteration)** *By repeating iteration of the diverse policy evaluation and the diverse policy improvement, any initial policy converges to the optimal policy $\pi^*$ s.t. $Q^{\pi^*}(s_t, a_t) \geq Q^{\pi'}(s_t, a_t), \forall \pi' \in \Pi, \forall (s_t, a_t) \in \mathcal{S} \times \mathcal{A}$. Also, such $\pi^*$ achieves maximum $J$, i.e., $J_{\pi^*}(\pi^*) \geq J_\pi(\pi)$ for any $\pi \in \Pi$.*

*Proof.* Let $\{\pi_i : i \geq 0, \pi_i \in \Pi\}$ be a sequence of policies s.t. $\pi_{i+1} = \arg\max_{\pi \in \Pi} J_{\pi_i}(\pi)$. For arbitrary state action pairs $(s, a) \in \mathcal{S} \times \mathcal{A}$, $\{Q^{\pi_i}(s, a)\}$ monotonically increases by Lemma 2 and each $Q^{\pi_i}(s, a)$ is bounded. Also, $\pi_{i+1}$ is obtained by the policy improvement that maximizes $J_{\pi_i}(\pi(\cdot|s))$, so $J_{\pi_i}(\pi_{i+1}(\cdot|s)) \geq J_{\pi_i}(\pi_i(\cdot|s))$ as stated in the proof of Lemma 2. From the definition of $J_{\pi_{old}}(\pi)$ in (8), all terms are the same for $J_{\pi_{i+1}}(\pi_{i+1}(\cdot|s))$ and $J_{\pi_i}(\pi_{i+1}(\cdot|s))$ except $\beta\mathbb{E}_{a\sim\pi_{i+1}}[Q^{\pi_{i+1}}(s, a)]$ in $J_{\pi_{i+1}}(\pi_{i+1}(\cdot|s))$ and $\beta\mathbb{E}_{a\sim\pi_{i+1}}[Q^{\pi_i}(s, a)]$ in $J_{\pi_i}(\pi_{i+1}(\cdot|s))$. Since $\{Q^{\pi_i}(s, a)\}$ monotonically increases, $J_{\pi_{i+1}}(\pi_{i+1}(\cdot|s)) \geq J_{\pi_i}(\pi_{i+1}(\cdot|s))$. Finally, $J_{\pi_{i+1}}(\pi_{i+1}(\cdot|s)) \geq J_{\pi_i}(\pi_{i+1}(\cdot|s)) \geq J_{\pi_i}(\pi_i(\cdot|s))$ for any state $s \in \mathcal{S}$, so the sequence $\{J_{\pi_i}(\pi_i(\cdot|s))\}$ also monotonically increases, and each $J_{\pi_i}(\pi_i(\cdot|s))$ is bounded because $Q$-function and the target entropy are bounded.

By the monotone convergence theorem, $\{Q^{\pi_i}\}$ and $\{J_{\pi_i}(\pi_i)\}$ pointwisely converge to their optimal functions $Q^* : \mathcal{S} \times \mathcal{A} \to \mathbb{R}$ and $J^* : \mathcal{S} \to \mathbb{R}$, respectively. Here, note that $J^*(s) \geq J_{\pi_i}(\pi_i(\cdot|s))$ for any $i$ because the sequence $\{J_{\pi_i}(\pi_i)\}$ is monotonically increasing. From the definition of convergent sequence, for arbitrary $\epsilon > 0$, there is a large $N \geq 0$ s.t. $J_{\pi_i}(\pi_i(\cdot|s)) \geq J^*(s) - \frac{\epsilon(1-\gamma)}{\gamma}$ satisfies for all $i \geq N$ and any $s \in \mathcal{S}$.

Now, we can easily show that $J_{\pi_k}(\pi_k(\cdot|s)) \geq J_{\pi_k}(\pi(\cdot|s)) - \frac{\epsilon(1-\gamma)}{\gamma}$ for any $k > N$, any policy $\pi \in \Pi$, and any $s \in \mathcal{S}$. (If not, $J_{\pi_k}(\pi_{k+1}) = \max_{\pi'} J_{\pi_k}(\pi') \geq J_{\pi_k}(\pi)$, and then $J_{\pi_{k+1}}(\pi_{k+1}(\cdot|s')) \geq J_{\pi_k}(\pi_{k+1}(\cdot|s')) \geq J_{\pi_k}(\pi(\cdot|s')) > J_{\pi_k}(\pi_k(\cdot|s')) + \frac{\epsilon(1-\gamma)}{\gamma} \geq J^*(s')$ for some $s' \in \mathcal{S}$. Clearly, it contradicts the monotone increase of the sequence $\{J_{\pi_i}(\pi_i)\}$.)

Then, by the similar way with (A.3),

$$Q^{\pi_k}(s_t, a_t) = \frac{1}{\beta}r_t + \gamma\mathbb{E}_{s_{t+1}\sim P}[V^{\pi_k}(s_{t+1})] = \frac{1}{\beta}r_t + \gamma\mathbb{E}_{s_{t+1}\sim P}[J_{\pi_k}(\pi_k(\cdot|s_{t+1}))]$$

$$\geq \frac{1}{\beta}r_t + \gamma\mathbb{E}_{s_{t+1}\sim P}\left[J_{\pi_k}(\pi(\cdot|s_{t+1})) - \frac{\epsilon(1-\gamma)}{\gamma}\right]$$

$$= \frac{1}{\beta}r_t + \gamma\mathbb{E}_{s_{t+1}\sim P}[\mathbb{E}_{a_{t+1}\sim\pi}[Q^{\pi_k}(s_{t+1}, a_{t+1}) + \alpha\log R^{\pi,\alpha}(s_{t+1}, a_{t+1}) - \alpha\log\alpha\pi(a_{t+1}|s_{t+1})]$$

$$+ (1-\alpha)\mathbb{E}_{a_{t+1}\sim q}[\log(1 - R^{\pi,\alpha}(s_{t+1}, a_{t+1})) - \log(1-\alpha)q(a_{t+1}|s_{t+1})]] - \epsilon(1-\gamma)$$

$$\vdots$$

$$\geq Q^\pi(s_t, a_t) - \epsilon. \tag{A.4}$$

Note that the state action pair $(s, a)$, the policy $\pi$, and $\epsilon > 0$ were arbitrary, so we can conclude that $Q^{\pi_\infty}(s, a) \geq Q^\pi(s, a)$ for any $\pi \in \Pi$ and $(s, a) \in \mathcal{S} \times \mathcal{A}$. In addition, we show that $J_{\pi_k}(\pi_k(\cdot|s)) \geq J_{\pi_k}(\pi(\cdot|s)) - \frac{\epsilon(1-\gamma)}{\gamma}$, so $J_{\pi_\infty}(\pi_\infty(\cdot|s)) \geq J_\pi(\pi(\cdot|s))$ for any $\pi \in \Pi$ and any $s \in \mathcal{S}$. Thus, $\pi_\infty$ is the optimal policy $\pi^*$, and we can conclude that $\{\pi_i\}$ converges to the optimal policy $\pi^*$.

## A.2 PROOF OF THEOREM 2

**Theorem 2** *Suppose that the policy is parameterized with parameter $\theta$. Then, for parameterized policy $\pi_\theta$, two objective functions $J_{\pi_{\theta_{old}}}(\pi_\theta(\cdot|s_t))$ and $\tilde{J}_{\pi_{\theta_{old}}}(\pi_\theta(\cdot|s_t))$ have the same gradient direction for $\theta$ at $\theta = \theta_{old}$ for all $s_t \in \mathcal{S}$.*

*Proof.* Under the parameterization of $\pi_\theta$, the two objective functions become

$$J_{\pi_{\theta_{old}}}(\pi_\theta(\cdot|s_t)) = \beta(\mathbb{E}_{a_t\sim\pi_\theta}[Q^{\pi_{\theta_{old}}}(s_t, a_t) + \alpha\log R^{\pi_\theta,\alpha}(s_t, a_t) - \alpha\log\pi_\theta(a_t|s_t)]$$

$$+ (1-\alpha)\mathbb{E}_{a_t\sim q}[\log(1 - R^{\pi_\theta,\alpha}(s_t, a_t)) - \log q(a_t|s_t)]) + H(\alpha)$$

$$\tilde{J}_{\pi_{\theta_{old}}}(\pi_\theta(\cdot|s_t)) = \beta\mathbb{E}_{a_t\sim\pi_\theta}[Q^{\pi_{old}}(s_t, a_t) + \alpha\log R^{\pi_{old},\alpha}(s_t, a_t) - \alpha\log\pi_\theta(a_t|s_t)].$$

We can ignore the common $Q$-function and $\log \pi_\theta$ terms, and the constant terms w.r.t. $\theta$ that leads zero gradient in both objective functions. Thus, we only need to show

$$\nabla_\theta[\alpha\mathbb{E}_{a_t\sim\pi_\theta}[\log R^{\pi_\theta,\alpha}] + (1-\alpha)\mathbb{E}_{a_t\sim q}[\log(1-R^{\pi_\theta,\alpha})]] = \nabla_\theta\mathbb{E}_{a_t\sim\pi_\theta}[\alpha\log R^{\pi_{\theta_{old}},\alpha}] \quad (A.5)$$

at $\theta = \theta_{old}$. Now, the gradient of the left term in (A.5) at $\theta = \theta_{old}$ can be expressed as

$$\begin{aligned}
&\nabla_\theta[\alpha\mathbb{E}_{a_t\sim\pi_\theta}[\log R^{\pi_\theta,\alpha}] + (1-\alpha)\mathbb{E}_{a_t\sim q}[\log(1-R^{\pi_\theta,\alpha})]] \\
&= \alpha\mathbb{E}_{a_t\sim\pi_\theta}[\log R^{\pi_\theta,\alpha} \cdot \nabla_\theta\log\pi_\theta] \\
&\quad + \alpha\mathbb{E}_{a_t\sim\pi_\theta}[\nabla_\theta\log R^{\pi_\theta,\alpha}] + (1-\alpha)\mathbb{E}_{a_t\sim q}[\nabla_\theta\log(1-R^{\pi_\theta,\alpha})] \\
&= \nabla_\theta\alpha\mathbb{E}_{a_t\sim\pi_\theta}[\alpha\log R^{\pi_{\theta_{old}},\alpha}] \\
&\quad + \alpha\mathbb{E}_{a_t\sim\pi_\theta}[\nabla_\theta\log R^{\pi_\theta,\alpha}] + (1-\alpha)\mathbb{E}_{a_t\sim q}[\nabla_\theta\log(1-R^{\pi_\theta,\alpha})]. \quad (A.6)
\end{aligned}$$

Here, the gradient of the last two terms in (A.6) becomes zero as shown below:

$$\begin{aligned}
&\alpha\mathbb{E}_{a_t\sim\pi_\theta}[\nabla_\theta\log R^{\pi_\theta,\alpha}] + (1-\alpha)\mathbb{E}_{a_t\sim q}[\nabla_\theta\log(1-R^{\pi_\theta,\alpha})] \\
&= \alpha\mathbb{E}_{a_t\sim\pi_\theta}[\nabla_\theta R^{\pi_\theta,\alpha}/R^{\pi_\theta,\alpha}] + (1-\alpha)\mathbb{E}_{a_t\sim q}[\nabla_\theta(1-R^{\pi_\theta,\alpha})/(1-R^{\pi_\theta,\alpha})] \\
&= \alpha\mathbb{E}_{a_t\sim\pi_\theta}[\nabla_\theta R^{\pi_\theta,\alpha}/R^{\pi_\theta,\alpha}] - (1-\alpha)\mathbb{E}_{a_t\sim q}[\nabla_\theta R^{\pi_\theta,\alpha}/(1-R^{\pi_\theta,\alpha})] \\
&= \alpha\mathbb{E}_{a_t\sim\pi_\theta}[\nabla_\theta R^{\pi_\theta,\alpha}/R^{\pi_\theta,\alpha}] - (1-\alpha)\mathbb{E}_{a_t\sim q}\left[\frac{\alpha\pi_\theta + (1-\alpha)q}{(1-\alpha)q} \cdot \nabla_\theta R^{\pi_\theta,\alpha}\right] \\
&\overset{(1)}{=} \alpha\mathbb{E}_{a_t\sim\pi_\theta}[\nabla_\theta R^{\pi_\theta,\alpha}/R^{\pi_\theta,\alpha}] - \alpha\mathbb{E}_{a_t\sim\pi_\theta}\left[\frac{\alpha\pi_\theta + (1-\alpha)q}{\alpha\pi_\theta} \cdot \nabla_\theta R^{\pi_\theta,\alpha}\right] \\
&= \alpha\mathbb{E}_{a_t\sim\pi_\theta}[\nabla_\theta R^{\pi_\theta,\alpha}/R^{\pi_\theta,\alpha}] - \alpha\mathbb{E}_{a_t\sim\pi_\theta}[\nabla_\theta R^{\pi_\theta,\alpha}/R^{\pi_\theta,\alpha}] = 0, \quad (A.7)
\end{aligned}$$

where we used an importance sampling technique at $\mathbb{E}_{a_t\sim q}[f(s_t,a_t)] = \mathbb{E}_{a_t\sim\pi_\theta}\left[\frac{q(a_t|s_t)}{\pi_\theta(a_t|s_t)}f(s_t,a_t)\right]$ for Step (1). By (A.6) and (A.7), $J_{\pi_{\theta_{old}}}(\pi_\theta(\cdot|s_t))$ and $J_{\pi_{\theta_{old}}}(\pi_\theta(\cdot|s_t))$ have the same gradient at $\theta = \theta_{old}$.

## B   DETAILED DAC IMPLEMENTATION

To compute the final objective function (9), we need to estimate $Q^{\pi_{old}}$ and $R^{\pi_{old},\alpha}$. $Q^{\pi_{old}}$ can be estimated by diverse policy evaluation. For estimation of $R^{\pi_{old},\alpha}$, we use function $R^\alpha$. If we set the objective function of the ratio function as $J(R^\alpha(s_t, \cdot)) = \alpha\mathbb{E}_{a_t \sim \pi}[\log R^\alpha(s_t, a_t)] + (1 - \alpha)\mathbb{E}_{a_t \sim q}[\log(1 - R^\alpha(s_t, a_t))]$. In the $\alpha = 0.5$ case, Generative Adversarial Network (GAN) (Goodfellow et al., 2014) has shown that the ratio function for $\alpha = 0.5$ can be estimated by maximizing $J(R^{0.5})$. By a similar way, we can easily show that maximizing $J(R^\alpha)$ can estimate our ratio function as below:

For given $s$, $J(R^\alpha(s, \cdot)) = \int_a \alpha\pi(a|s)\log R^\alpha(s, a) + (1 - \alpha)q(a|s)\log(1 - R^\alpha(s, a))da$. The integrand is in the form of $y \to a\log y + b\log(1 - y)$ with $a = \alpha\pi$ and $b = (1 - \alpha)q$. For any $(a, b) \in \mathbb{R}^2\backslash(0, 0)$, the function $y \to a\log y + b\log(1 - y)$ has its maximum at $a/(a+b)$. Thus, the optimal $R^{*,\alpha}$ maximizing $J(R^\alpha(s, \cdot))$ is $R^{*,\alpha}(s, a) = \alpha\pi/(\alpha\pi + (1 - \alpha)q) = R^{\pi,\alpha}(s_t, a_t)$. Here, note that $J(R^\alpha)$ becomes just an $\alpha$-skewed Jensen-Shannon (JS) divergence except some constant terms if $R^\alpha = R^{\pi,\alpha}$.

For implementation we use deep neural networks to approximate the policy $\pi$, the diverse value functions $Q$, $V$, and the ratio function $R^\alpha$, and their network parameters are given by $\theta$, $\phi$, $\psi$, and $\eta$, respectively. Based on Section 4.3 and we provide the practical objective (or loss) functions for parameter update as $\hat{J}_\pi(\theta)$, $\hat{J}_{R^\alpha}(\eta)$, $\hat{L}_Q(\phi)$, and $\hat{L}_V(\psi)$. The objective functions for the policy $\pi$ and the ratio function $R^\alpha$ are respectively given by

$$\hat{J}_\pi(\theta) = \mathbb{E}_{s_t \sim \mathcal{D},\, a_t \sim \pi_\theta}[Q_\phi(s_t, a_t) + \alpha\log R^\alpha_\eta(s_t, a_t) - \alpha\log\pi_\theta(a_t|s_t)], \tag{B.1}$$

$$\hat{J}_{R^\alpha}(\eta) = \mathbb{E}_{s_t \sim \mathcal{D}}[\alpha\mathbb{E}_{a_t \sim \pi_\theta}[\log R^\alpha_\eta(s_t, a_t)] + (1 - \alpha)\mathbb{E}_{a_t \sim \mathcal{D}}[\log(1 - R^\alpha_\eta(s_t, a_t))]]. \tag{B.2}$$

Furthermore, based on the Bellman operator, the loss functions for the value functions $Q$ and $V$ are given respectively given by

$$\hat{L}_Q(\phi) = \mathbb{E}_{(s_t,\, a_t) \sim \mathcal{D}}\left[\frac{1}{2}(Q_\phi(s_t, a_t) - \hat{Q}(s_t, a_t))^2\right], \tag{B.3}$$

$$\hat{L}_V(\psi) = \mathbb{E}_{s_t \sim \mathcal{D}}\left[\frac{1}{2}(V_\psi(s_t) - \hat{V}(s_t))^2\right], \tag{B.4}$$

where the target values are defined as

$$\hat{Q}(s_t, a_t) = \frac{1}{\beta}r_t + \gamma\mathbb{E}_{s_{t+1} \sim P}[V_{\bar{\psi}}(s_{t+1})] \tag{B.5}$$

$$\hat{V}(s_t) = \mathbb{E}_{a_t \sim \pi_\theta}[Q_\phi(s_t, a_t) + \alpha\log R^\alpha_\eta(s_t, a_t) - \alpha\log\alpha\pi_\theta(a_t|s_t)]$$
$$+ (1 - \alpha)\mathbb{E}_{a_t \sim \mathcal{D}}[\log(1 - R^\alpha_\eta(s_t, a_t)) - \log(1 - \alpha)q(a_t|s_t)]. \tag{B.6}$$

By using the property of ratio function that satisfies $\log(1 - R^{\pi,\alpha}) - \log(1 - \alpha)q = -\log(\alpha\pi + (1 - \alpha)q) = \log R^{\pi,\alpha} - \log\alpha\pi$, we can replace the last term in $\hat{V}(s_t)$ as $(1 - \alpha)\mathbb{E}_{a_t \sim \mathcal{D}}[\log R^\alpha_\eta(s_t, a_t) - \log\alpha\pi(a_t|s_t)]$. However, the probability of $\pi$ for actions sampled from $\mathcal{D}$ can have high variance, so we clip the term in the expectation over $a_t \sim \mathcal{D}$ by action dimension for stable learning, then the final target value becomes

$$\hat{V}(s_t) = \mathbb{E}_{a_t \sim \pi_\theta}[Q_\phi(s_t, a_t) + \alpha\log R^\alpha_\eta(s_t, a_t) - \alpha\log\alpha\pi_\theta(a_t|s_t)]$$
$$+ (1 - \alpha)\mathbb{E}_{a_t \sim \mathcal{D}}[\text{clip}(\log R^\alpha_\eta(s_t, a_t) - \log\alpha\pi(a_t|s_t), -d, d)], \tag{B.7}$$

where $d = \dim(\mathcal{A})$ is the action dimension. We will use (B.7) for implementation. Then, note that all objective (or loss) functions does not require the explicit $q$, and they can be represented by using the ratio function $R^\alpha$ only as explained in Section 4.3.

In addition, $R^\alpha \in (0, 1)$ should be guaranteed in the proof of Theorem 1, and $R^\alpha \in (0, 1)$ satisfies when $\pi$ and $q$ are non-zero for all state-action pairs. For practical implementation, we clipped the ratio function as $(\epsilon, 1 - \epsilon)$ for small $\epsilon > 0$ since some $q$ values can be close to zero before the replay buffer stores a sufficient amount of samples. $\pi$ is always non-zero since we consider Gaussian policy.

Here, $\bar{\psi}$ is the network parameter of the target value $V_{\bar{\psi}}$ updated by exponential moving average (EMA) of $\psi$ for stable learning (Mnih et al., 2015). Combining all up to now, we propose the

diversity actor-critic (DAC) algorithm summarized as Algorithm 1 in Appendix C. Note that DAC becomes SAC when $\alpha = 1$, and becomes standard off-policy RL without entropy regularization when $\alpha = 0$.

To compute the gradient of $\hat{J}_\pi(\theta)$, we use the reparameterization trick proposed by (Kingma & Welling, 2013; Haarnoja et al., 2018a). Note that the policy action $a_t \sim \pi_\theta$ is the output of the policy neural network with parameter $\theta$. So, it can be viewed as $a_t = f_\theta(\epsilon_t; s_t)$, where $f$ is a function parameterized by $\theta$ and $\epsilon_t$ is a noise vector sampled from spherical normal distribution $\mathcal{N}$. Then, the gradient of $\hat{J}_\pi(\theta)$ is represented as $\nabla_\theta \hat{J}_\pi(\theta) = \mathbb{E}_{s_t \sim \mathcal{D}, \epsilon_t \sim \mathcal{N}}[\nabla_a(Q_\phi(s_t, a) + \alpha \log R_\eta^\alpha(s_t, a) - \alpha \log \pi_\theta(a|s_t))|_{a=f_\theta(\epsilon_t; s_t)} \nabla_\theta f_\theta(\epsilon_t; s_t) - \alpha(\nabla_\theta \log \pi_\theta)(f_\theta(\epsilon_t; s_t)|s_t)]$.

For implementation, we use two $Q$-functions $Q_{\phi_i}$, $i = 1, 2$ to reduce overestimation bias as proposed in (Fujimoto et al., 2018), and each Q-function is updated to minimize their loss function $\hat{L}_Q(\phi_i)$. For the policy and the value function update, the minimum of two $Q$-functions is used (Haarnoja et al., 2018a).

Note that one version of SAC (Haarnoja et al., 2018b) considers adaptation of the entropy control factor $\beta$ by using the Lagrangian method with constraint $\mathcal{H}(\pi) \geq c$. In our case, this approach can also be generalized, but it is beyond the scope of the current paper and we only consider fixed $\beta$ in this paper.

## B.1 DETAILED IMPLEMENTATION OF THE $\alpha$-ADAPTATION

In order to learn $\alpha$, we parameterize $\alpha$ as a function of $s_t$ using parameter $\xi$, i.e., $\alpha = \alpha_\xi(s_t)$, and implement $\alpha_\xi(s_t)$ with a neural network. Then, $\xi$ is updated to minimize the following loss function deduced from (10):

$$\hat{L}_\alpha(\xi) = \mathbb{E}_{s_t \sim \mathcal{D}}[\alpha_\xi \mathcal{H}(\pi_\theta) + D_{JS}^{\alpha_\xi}(\pi_\theta || q) + (1 - \alpha_\xi)\mathcal{H}(q) - \alpha_\xi c] \tag{B.8}$$

Here, all the updates for diverse policy iteration is the same except that $\alpha$ is replaced with $\alpha_\xi(s_t)$. Then, the gradient of $\hat{L}_\alpha(\xi)$ with respect to $\xi$ can be estimated as below:

The loss function of $\alpha$ is defined as $\hat{L}_\alpha(\xi) = \mathbb{E}_{s_t \sim \mathcal{D}}[\alpha_\xi \mathcal{H}(\pi_\theta) + D_{JS}^{\alpha_\xi}(\pi_\theta || q) + (1 - \alpha_\xi)\mathcal{H}(q) - \alpha_\xi c]$. The gradient of $\hat{L}_\alpha(\xi)$ can be computed as

$$\nabla_\xi \hat{L}_\alpha(\xi) = \nabla_\xi \mathbb{E}_{s_t \sim \mathcal{D}}[\alpha_\xi \mathcal{H}(\pi_\theta) + D_{JS}^{\alpha_\xi}(\pi_\theta || q) + (1 - \alpha_\xi)\mathcal{H}(q) - \alpha_\xi c]$$
$$= \nabla_\xi \mathbb{E}_{s_t \sim \mathcal{D}}[\alpha_\xi \mathbb{E}_{a_t \sim \pi_\theta}[-\log(\alpha_\xi \pi_\theta + (1 - \alpha_\xi)q) - c] + (1 - \alpha_\xi)\mathbb{E}_{a_t \sim q}[-\log(\alpha_\xi \pi_\theta + (1 - \alpha_\xi)q)]]$$
$$= \mathbb{E}_{s_t \sim \mathcal{D}}[(\nabla_\xi \alpha_\xi)(\mathbb{E}_{a_t \sim \pi_\theta}[-\log(\alpha_\xi \pi_\theta + (1 - \alpha_\xi)q) - c] - \mathbb{E}_{a_t \sim q}[-\log(\alpha_\xi \pi_\theta + (1 - \alpha_\xi)q)])]$$
$$\quad + \mathbb{E}_{s_t \sim \mathcal{D}}[\alpha_\xi \mathbb{E}_{a_t \sim \pi_\theta}[-\nabla_\xi \log(\alpha_\xi \pi_\theta + (1 - \alpha_\xi)q)] + (1 - \alpha_\xi)\mathbb{E}_{a_t \sim q}[-\nabla_\xi \log(\alpha_\xi \pi_\theta + (1 - \alpha_\xi)q)]]$$
$$= \mathbb{E}_{s_t \sim \mathcal{D}}[(\nabla_\xi \alpha_\xi)(\mathbb{E}_{a_t \sim \pi_\theta}[-\log \alpha_\xi \pi_\theta + \log R^{\pi_\theta, \alpha_\xi} - c] - \mathbb{E}_{a_t \sim q}[\log(1 - R^{\pi_\theta, \alpha_\xi}) - \log(1 - \alpha_\xi)q])]$$
$$\quad + \mathbb{E}_{s_t \sim \mathcal{D}}\left[\underbrace{\int_{a_t \in \mathcal{A}} (\alpha_\xi \pi_\theta + (1 - \alpha_\xi)q)[-\nabla_\xi \log(\alpha_\xi \pi_\theta + (1 - \alpha_\xi)q)]}_{=0}\right]$$
$$= \mathbb{E}_{s_t \sim \mathcal{D}}[(\nabla_\xi \alpha_\xi)(\mathbb{E}_{a_t \sim \pi_\theta}[-\log \alpha_\xi \pi_\theta + \log R^{\pi_\theta, \alpha_\xi} - c] - \mathbb{E}_{a_t \sim q}[\log R^{\pi_\theta, \alpha_\xi} - \log \alpha_\xi \pi_\theta])] \tag{B.9}$$

Note that $R^{\pi_\theta, \alpha_\xi}$ can be estimated by the ratio function $R_\eta^{\alpha_\xi}$. Here, we use the same clipping technique as used in (B.7) for the last term of (B.9).

## C  ALGORITHM

---

**Algorithm 1** Diversity Actor Critic

---

Initialize parameter $\theta, \eta, \psi, \bar{\psi}, \xi, \phi_i, \ i = 1, 2$
**for** each iteration **do**
  Sample a trajectory $\tau$ of length $N$ by using $\pi_\theta$
  Store the trajectory $\tau$ in the buffer $\mathcal{D}$
  **for** each gradient step **do**
    Sample random minibatch of size $M$ from $\mathcal{D}$
    Compute $\hat{J}_\pi(\theta), \hat{J}_{R^\alpha}(\eta), \hat{L}_Q(\phi_i), \hat{L}_V(\psi)$ from the minibatch
    $\theta \leftarrow \theta + \delta \nabla_\theta \hat{J}_\pi(\theta)$
    $\eta \leftarrow \eta + \delta \nabla_\eta \hat{J}_{R^\alpha}(\eta)$
    $\phi_i \leftarrow \phi_i - \delta \nabla_{\phi_i} \hat{L}_Q(\phi_i), \ i = 1, 2$
    $\psi \leftarrow \psi - \delta \nabla_\psi \hat{L}_V(\psi)$
    Update $\bar{\psi}$ by EMA from $\psi$
    **if** $\alpha$-Adpatation **then**
      Compute $\hat{L}_\alpha(\xi)$ from the minibatch
      $\xi \leftarrow \xi - \delta \nabla_\xi \hat{L}_\alpha(\xi)$
    **end if**
  **end for**
**end for**

---

## D  HYPERPARAMETER SETUP AND ENVIRONMENT DESCRIPTION

In Table D.1, we provide the detailed hyperparameter setup for DAC and the SAC baselines: SAC, and SAC-Div. Table D.2 shows the environment description, the corresponding entropy control coefficient $\beta$, threshold for sparse Mujoco tasks, and reward delay $D$ for delayed Mujoco tasks.

| | SAC / SAC-Div | DAC |
|---|---|---|
| Learning rate $\delta$ | $3 \cdot 10^{-4}$ | |
| Discount factor $\gamma$ | 0.99 (0.999 for pure exploration) | |
| Horizon $N$ | 1000 | |
| Mini-batch size $M$ | 256 | |
| Replay buffer length | $10^6$ | |
| Smoothing coefficient of EMA for $V_{\bar{\psi}}$ | 0.005 | |
| Optimizer | Adam | |
| Num. of hidden layers (all networks) | 2 | |
| Size of hidden layers (all networks) | 256 | |
| Policy distribution | Independent Gaussian distribution | |
| Activation layer | ReLu | |
| Output layer for $\pi_\theta, Q_\phi, V_\psi, V_{\bar{\psi}}$ | Linear | |
| Output layer for $\alpha_\xi, R_\eta^\alpha$ | $\cdot$ | Sigmoid |
| Regularize coefficient for $\alpha_\xi$ | $\cdot$ | $10^{-3}$ |
| Control coefficient $c$ for $\alpha$-adaptation | $\cdot$ | $-\dim(\mathcal{A})$ |

Table D.1: Hyperparamter setup

|  | State dim. | Action dim. | $\beta$ | Threshold |
|---|---|---|---|---|
| SparseHalfCheetah-v1 | 17 | 6 | 0.02 | 5.0 |
| SparseHopper-v1 | 11 | 3 | 0.04 | 1.0 |
| SparseWalker2d-v1 | 17 | 6 | 0.02 | 1.0 |
| SparseAnt-v1 | 111 | 8 | 0.02 | 1.0 |
|  | State dim. | Action dim. | $\beta$ | Delay $D$ |
| HumanoidStandup-v1 | 376 | 17 | 1 | . |
| DelayedHalfCheetah-v1 | 17 | 6 | 0.2 | 20 |
| DelayedHopper-v1 | 11 | 3 | 0.2 | 20 |
| DelayedWalker2d-v1 | 17 | 6 | 0.2 | 20 |
| DelayedAnt-v1 | 111 | 8 | 0.2 | 20 |

Table D.2: State and action dimensions of Mujoco tasks and the corresponding $\beta$

## E    SIMULATION SETUP

We compared our DAC algorithm with the SAC baselines and other RL algorithms on various types of Mujoco tasks with continuous action spaces (Todorov et al., 2012) in OpenAI GYM (Brockman et al., 2016). For fairness, both SAC/SAC-Div and DAC used a common hyperparameter setup that basically follows the setup in (Haarnoja et al., 2018a). Detailed hyperparameter setup and environment description are provided in Appendix D, and the entropy coefficient $\beta$ is selected based on the ablation study in Section 6.3. For the policy space $\Pi$ we considered Gaussian policy set widely considered in usual continuous RL. For the performance plots in this section, we used deterministic evaluation which generated an episode by deterministic policy for each iteration, and the shaded region in the figure represents standard deviation ($1\sigma$) from the mean.

## F    PERFORMANCE COMPARISONS

In this section, we provide more performance plots and tables. In Section F.1, Fig. F.1 shows the mean number of discretized state visitation curve of DAC and SAC/SAC-Div. For discretization, we simply consider 2 components of observations of Mujoco tasks, which indicate the position of the agent: $x, z$ axis position for SparseHalfCheetah, SparseHopper, and SparseWalker, and $x, y$ axis position for SparseAnt. We discretize the position by setting the grid spacing per axis to 0.01 in range $(-10, 10)$. Table. F.1 shows the performance on sparse Mujoco tasks. Table F.2 shows max average return for HumanoidStandup and delayed Mujoco tasks. In Section F.3, Fig. F.3 and Table. F.3 shows the performance comparison to other RL algorithms on HumanoidStandup and delayed Mujoco tasks.

### F.1    PERFORMANCE COMPARISON WITH THE SAC BASELINES

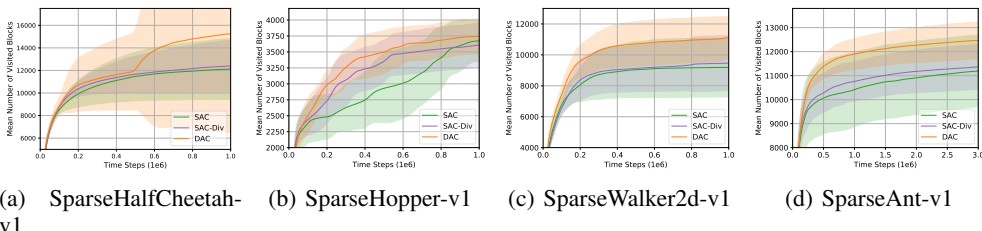

(a)    SparseHalfCheetah-v1  (b)  SparseHopper-v1  (c)  SparseWalker2d-v1  (d)  SparseAnt-v1

Figure F.1: The number of discretized state visitation on sparse Mujoco tasks

|  | DAC ($\alpha = 0.5$) | SAC | SAC-Div |
|---|---|---|---|
| SparseHalfCheetah | **915.90±50.71** | 386.90±404.70 | 394.70±405.53 |
| SparseHopper | 896.90±10.57 | 900.60±5.22 | **901.40±4.25** |
| SparseWalker2d | **573.10±404.96** | 301.30±408.15 | 373.10±433.13 |
| SparseAnt | 935.80±37.08 | 870.70±121.14 | **963.80±42.51** |

Table F.1: Max average return of DAC algorithm and SAC baselines for fixed $\alpha$ setup

|  | DAC ($\alpha = 0.5$) | DAC ($\alpha = 0.8$) | DAC ($\alpha$-adapt.) | SAC | SAC-Div |
|---|---|---|---|---|---|
| HumanoidS | **202491.81** ±**25222.77** | 170832.05 ±12344.71 | 197302.37 ±43055.31 | 167394.36 ±7291.99 | 165548.76 ±2005.85 |
| Del. HalfCheetah | 6071.93±1045.64 | 6552.06±1140.18 | **7594.70±1259.23** | 3742.33±3064.55 | 4080.67±3418.07 |
| Del. Hopper | 3283.77±112.04 | 2836.81±679.05 | **3428.18±69.08** | 2175.31±1358.39 | 2090.64±1383.83 |
| Del. Walker2d | **4360.43±507.58** | 3973.37±273.63 | 4067.11±257.81 | 3220.92±1107.91 | 4048.11±290.48 |
| Del. Ant | 4088.12±578.99 | 3535.72±1164.76 | **4243.19±795.49** | 3248.43±1454.48 | 3978.34±1370.23 |

Table F.2: Max average return of DAC algorithms and SAC baselines for adaptive $\alpha$ setup

## F.2 COMPARISON TO RND AND MAXENT

We first compared the pure exploration performance of DAC to random network distillation (RND) (Burda et al., 2018) and MaxEnt (Hazan et al., 2019), which are state-of-the-art exploration methods, on the continuous 4-room maze task described in Section 6.1. RND adds an intrinsic reward $r_{int,t}$ to MDP extrinsic reward $r_t$ as $r_{RND,t} = r_t + c_{int}r_{int,t}$ based on the model prediction error $r_{int,t} = ||\hat{f}(s_{t+1}) - f(s_{t+1})||^2$ of prediction network $\hat{f}$ and random target network $f$ for given state $s_{t+1}$. The parameter of the target network is initially given randomly and the prediction network learns to minimize the MSE of the two models. Then, the agent goes to rare states since rare states have higher prediction errors. On the other hand, MaxEnt considers maximizing the entropy of state mixture distribution $d^{\pi^{mix}}$ by setting the reward functional in (Hazan et al., 2019) as $-\log d^{\pi^{mix}}(s) + c_M$, where $d^\pi$ is a state distribution of the trajectory generated from $\pi$ and $c_M$ is a smoothing constant. Here, MaxEnt mainly considers large or continous state space, so the reward functional is computed based on several projection/discretization methods. Then, MaxEnt explores the state space better than a simple random policy on various tasks with continuous state space.

For RND, for both the prediction network and the target network, we use MLP with 2 ReLu hidden layers of size 256, where the input dimension is equal to the state dimension and the output dimension is 20, and we use $c_{int} = 1$. For MaxEnt, we compute the reward functional at each iteration by using Kernel density estimation with a bandwidth 0.1 as stated in (Hazan et al., 2019) on previous 10000 states stored in the buffer, and we use $c_M = 0.01$. For RND and MaxEnt, we change the entropy term of SAC/DAC to the intrinsic reward and the reward functional term respectively, and we use the Gaussian policy with fixed standard deviation $\sigma = 0.1$. Fig. F.2(a) shows the mean number of state visitation curve over 30 seeds of the 4-room maze task and Fig. F.2(b) shows the corresponding state visit histogram of all seeds. As seen in Fig. F.2, DAC explores more number of states than RND and MaxEnt on continuous 4-room maze task, so it is seen that the exploration of DAC is more sample-efficient than that of RND/MaxEnt on the maze task.

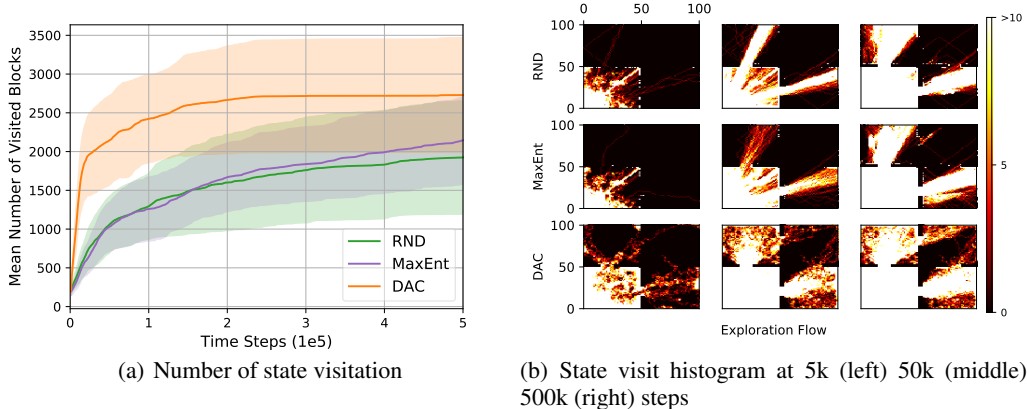

(a) Number of state visitation

(b) State visit histogram at 5k (left) 50k (middle) 500k (right) steps

Figure F.2: Pure exploration comparison with RND/MaxEnt

### F.3 COMPARISON TO OTHER RL ALGORITHMS

We also compare the performance of DAC with $\alpha$-adaptation to other state-of-the-art RL algorithms. Here, we consider various on-policy RL algorithms: Proximal Policy Optimization (Schulman et al., 2017b) (PPO, a stable and popular on-policy algorithm), Actor Critic using Kronecker-factored Trust Region (Wu et al., 2017) (ACKTR, actor-critic that approximates natural gradient by using Kronecker-factored curvature), and off-policy RL algorithms: Twin Delayed Deep Deterministic Policy Gradient (Fujimoto et al., 2018) (TD3, using clipped double-Q learning for reducing over-estimation); and Soft Q-Learning (Haarnoja et al., 2017) (SQL, energy based policy optimization using Stein variational gradient descent). We used implementations in OpenAI baselines (Dhariwal et al., 2017) for PPO and ACKTR, and implementations in author's Github for other algorithms. We provide the performance results as Fig. F.3 and Table F.3.

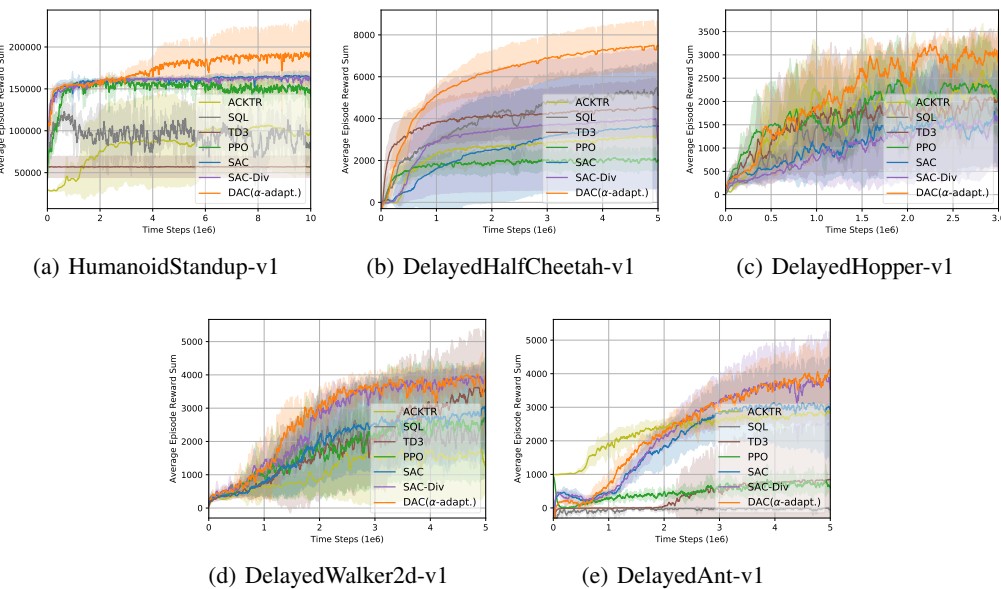

(a) HumanoidStandup-v1

(b) DelayedHalfCheetah-v1

(c) DelayedHopper-v1

(d) DelayedWalker2d-v1

(e) DelayedAnt-v1

Figure F.3: Performance comparison to other RL algorithms

|  | DAC | PPO | ACKTR | SQL | TD3 | SAC |
|---|---|---|---|---|---|---|
| HumanoidS | **197302.37** **±43055.31** | 160211.90 ±3268.37 | 109655.30 ±49166.15 | 138996.84 ±33903.03 | 58693.87 ±12269.93 | 167394.36 ±7291.99 |
| Del. HalfCheetah | **7594.70** **±1259.23** | 2247.92 ±640.69 | 3295.30 ±824.05 | 5673.34 ±1241.30 | 4639.85 ±1393.95 | 3742.33 ±3064.55 |
| Del. Hopper | **3428.18** **±69.08** | 2740.15 ±719.63 | 2864.81 ±1072.64 | 2720.32 ±127.71 | 2276.58 ±1471.66 | 2175.31 ±1358.39 |
| Del. Walker2d | **4067.11** **±257.81** | 2859.27 ±1938.50 | 1927.32 ±1647.49 | 3323.63 ±503.18 | 3736.72 ±1806.37 | 3220.92 ±1107.91 |
| Del. Ant | **4243.19** **±795.49** | 1224.33 ±521.62 | 2956.51 ±234.89 | 6.59 ±16.42 | 904.99 ±1811.78 | 3248.43 ±1454.48 |

Table F.3: Max average return of DAC and other RL algorithms

# G   ABLATION STUDIES

Here, we provide more ablation studies for remaining delayed Mujoco tasks. Fig. G.2 shows the averaged learning curves of $\alpha$, $D_{JS}^{\alpha}$, and $\mathcal{H}(\pi)$ of DAC considering $\alpha$-adaptation, where the control coefficient $c$ is $-2.0d$ and $d = \dim(\mathcal{A})$. Fig. G.2 shows the performance of DAC considering $\alpha$-adaptation with control coefficient $c = 0, -0.5d, -1.0d$, and $-2.0d$. Fig. G.3 shows the performance of DAC with $\alpha = 0.5$ and $\beta = 0.1, 0.2, 0.4$. Fig. G.4 shows the performance of SAC, SAC-Div with KL-divergence (SAC-Div(KL)), SAC-Div with JS-divergence (SAC-Div(JS)), and DAC to see the effect of JS divergence on the performance as explained in Section 6.3. Other hyperparameters follow the default setup given in Table D.1.

**Weighting factor $\alpha$**

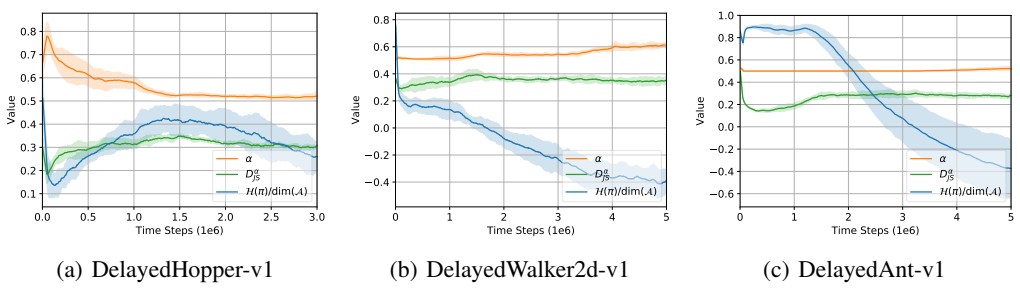

(a) DelayedHopper-v1          (b) DelayedWalker2d-v1          (c) DelayedAnt-v1

Figure G.1: Ablation study for $\alpha$

**Control coefficient $c$**

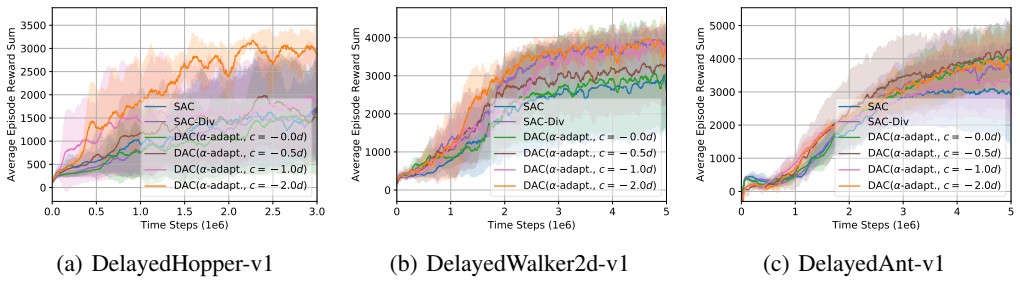

(a) DelayedHopper-v1          (b) DelayedWalker2d-v1          (c) DelayedAnt-v1

Figure G.2: Ablation study for $c$

**Entropy coefficient** $\beta$

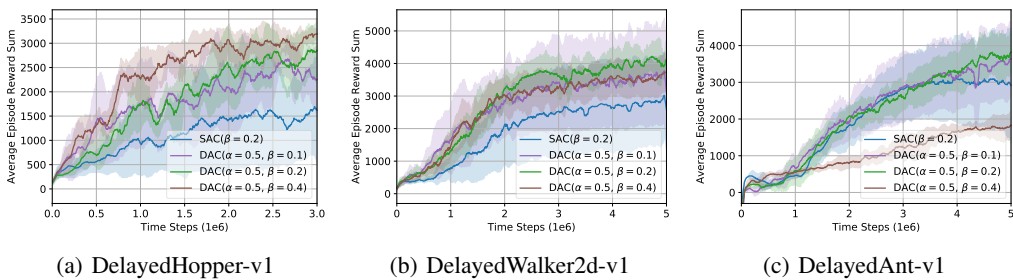

(a) DelayedHopper-v1    (b) DelayedWalker2d-v1    (c) DelayedAnt-v1

Figure G.3: Ablation study for $\beta$

**The Effect of JS divergence**

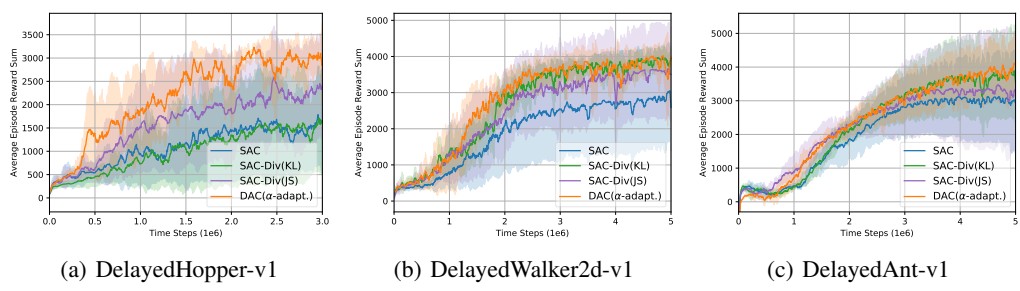

(a) DelayedHopper-v1    (b) DelayedWalker2d-v1    (c) DelayedAnt-v1

Figure G.4: Ablation study for JS divergence

