# OpenReview forum: "Diversity Actor-Critic: Sample-Aware Entropy Regularization for Sample-Efficient Exploration"
_ICLR.cc/2021/Conference — Reject_

### Official Review · AnonReviewer3 · 2020-10-28
**Interesting, but poorly motivated and positioned**

**Rating:** 5
**Confidence:** 4

**Review:**

### Summary

The paper proposes DAC, an actor-critic method exploiting the replay buffer to do policy entropy regularisation. The main idea of DAC is to use the data from the replay buffer to induce a distribution  $q(\cdot, s_t)$ and replace the entropy part of the Soft Actor-Critic objective with a convex combination of $q$ and $\pi$. This results positively on exploration properties and leads to sample-efficiency gains on some of the considered MuJoCo benchmarks.

### Pros
- Formulating the diversity using the entropy of the replay buffer frequences is an interesting idea.
- Using the convex combination of $q$ and $\pi$ for entropy regularisation is a nice way of generalising SAC for the considered purpose.
- The paper shows the convergence of their method to an optimal policy and derives a surrogate objective whose gradient direction coincides with the original one, but which can be practically used. (However, I have not checked the proofs which are in the appendix).

### Cons
- It is not clear, what is the problem the paper tackles. Is it exploration? Is it a generic RL setup? What kind of problems is DAC good for?
- If DAC is for improving exploration, then it should be compared with other exploration methods, not with vanilla SAC. Comparison with RND should not be in the appendix and there should be more details on this. Related work in this case should have a paragraph on exploration methods in RL.
- The paper is based on assumptions not challenged/tested by the authors, e.g. policy entropy regularisation is inefficient, because it does not take the distribution of the samples into account.
- The paper focuses more on the technical details of the solution rather than justifying the assumptions and making the research question clear.

### Reasoning behind the score

I believe, the paper has a great potential. However, at the moment I vote for rejection. The paper has to have a clear research question and its motivation. This should define the experimental part of the work. Lack of a clear positioning makes it unclear if the baselines of the experimental sections are the right ones and whether the claims have been properly supported by the results.

### Questions to the authors
- Can you formulate the exact problem you are solving?
- How can you justify the claim that 'entropy regularization is sample inefficient in off-policy learning since it does not take the distribution of previous samples stored in the replay buffer into account.
- "it is preferable that the old sample distribution in the replay buffer is uniformly distributed". Why is it true? Doesn't prioritized experience replay refute this claim?
- You define $\beta$ in Equation 1 in $(0, \infty)$, can it really be infinite?
- "The rationale behind this is that it is preferable to have as diverse actions stored in the replay buffer as possible for better Q estimation in off-policy learning." What are the assumptions for this? Do you care more about better Q estimates or finding an better policy faster? How can you support your rationale?
- In section 4.1. you define the target distribution as a convex combination of $\pi$ and $q$. You assume that the buffer is generated by $q$. Does such a policy always exist? What are the assumptions for this?
- You prove the convergence of your algorithm (I did not check the proof in the appendix), what are the assumptions for which the convergence is guaranteed?
- Why do you use sparse/delayed MuJoCo benchmarks, but not the original ones?
- The variance across different seeds seems to be huge for your method (as well as for the others). What do you think is the reason behind this? This also happens for the pure exploration task in 6.1, why do you think it happens?
- For the adaptive $\alpha$ case, you restrict the range of possible values, what is the reasoning behind the left boundary?
- I think your paper can find an important application in Imitation Learning or off-line RL. Have you considered this? Are you aware of works which do something similar in those subfields?

### Additional feedback not affecting the score
- "Reinforcement learning aims to maximize the discounted sum of rewards...'. Should be 'expected discounted sum'.
- There should be a distribution over initial states under the expectation sign in 3.1.
- 'A is the continuous action space'. This is not true for the general MDP definition, specify that this is specific for your paper.
- Section 3.1, a policy is a mapping from states to distribution over actions, not to actions.
- In off-policy, we can learn from any other samples, not only from 'previous samples' from our policy.
- typo "propoed" at the bottom of page 4.
- Equation 9 does not have a left hand side.
- DAC acronym has been used in RL. I would choose a different one to avoid confusion.

---

> ### Author Response · Authors · 2020-11-11
> **Reply for reviewer 3**
>
> Thank you for the valuable comments and constructive feedbacks. We provide our feedback below:
>
> •	It is not clear, what is the problem the paper tackles. ... What kind of problems is DAC good for?
>
> •	Can you formulate the exact problem you are solving?
>
> •	How can you justify the claim that 'entropy regularization is sample inefficient in off-policy learning since it does not take the distribution of previous samples stored in the replay buffer into account.
>
> •	"The rationale behind this is that it is preferable to have as diverse actions stored in the replay buffer as possible for better Q estimation in off-policy learning." What are the assumptions for this? Do you care more about better Q estimates or finding an better policy faster? How can you support your rationale?
>
> In order to guarantee the convergence of Q-learning [R3-1], there is a key assumption: Each state-action pair must be visited infinitely often. If the policy does not visit diverse state-action pairs many times, it converges to local optima. Therefore, exploration for visiting different state-action pairs is important for RL, and the original entropy regularization encourages exploration [R3-2].
>
> The simple policy entropy maximization will choose all actions with equal probability without considering the previous action samples. In contrast, if we consider maximizing the entropy of the mixture of $\pi$ and $q$ (it becomes the future sample distribution since the buffer stores samples generated by $\pi$), $\pi$ should choose actions rare in the buffer with high probability and actions stored many times in the buffer with low probability to make the mixture distribution uniform. Hence, it considers the samples already stored in the buffer in choosing current action and encourages sample-efficient exploration. We provide a simple example below:
>
> Let us consider a simple 1-step MDP in which $s_0$ is the unique initial state, there exist $N$ actions ($a_0\in(\{1,\cdots,N\})$), $s_1$ is the terminal state, and $r$ is the deterministic reward function. Then, there exist $N$ state-action pairs in total and let us assume that we already have $N-1$ state-action samples in the replay buffer as $R=(\{(s_0,1,r(s_0,1)),\cdots,(s_0,N-1,r(s_0,N-1))\})$. In order to estimate the Q-function for all state-action pairs, the policy should sample Action $N$ (After then, we can reuse all samples infinitely to estimate Q). Here, we will compare two exploration methods.
>
> 1)	First, if we consider the simple entropy maximization, the policy will choose all actions with equal probability $1/N$ (uniformly) since $\max_\pi \mathcal{H}(\pi) =\min_\pi KLD(\pi||U)$ is achieved when $\pi=U$, where $U$ is a uniform distribution. Then, $N$ samples should be taken on average by the policy to visit Action $N$.
>
> 2)	Consider the sample-aware entropy maximization as in our paper. Here, the sample action distribution $q$ in the buffer is defined as $q(a_0|s_0)=1/(N-1)$ for $a_0\in\{0,\cdots,N-1\}$, and $q(N|s_0)=0$.
>
> Now, we set the target sample distribution as the mixture of $\pi$ and $q$, as $q_{target}^{\pi,\alpha}=\alpha\pi+(1-\alpha) q$, and we set $\alpha=1/N$. Then, in order to maximize the entropy of the target sample distribution $\max_\pi \mathcal{H}( q_{target}^{\pi,\alpha} )  =  \min_\pi KLD( q_{target}^{\pi,\alpha} ||U)$ , the policy distribution should be $\pi(N|s_0)=1$ to make $q_{target}^{\pi,\alpha}$ uniform. Thus, it only needs one sample to visit Action $N$.
> In this way, the simple entropy regularization is sample-inefficient for off-policy RL, and the proposed sample-aware entropy regularization enhances the sample-efficiency for exploration by using the previous sample distribution and choosing proper $\alpha$. With this motivation, we propose the sample-aware entropy regularization and the corresponding $\alpha$-adaptation. Therefore, the method in this paper addresses sample-efficient exploration.
>
> •	"it is preferable that the old sample distribution in the replay buffer is uniformly distributed". Why is it true? Doesn't prioritized experience replay refute this claim?
>
> The claim means that the policy should choose samples to make the sample distribution uniform over the overall state-action space to visit all state-action pairs for convergence of Q [R3-1] as explained the first comment (I think that the word “old sample” is misleading).  Note that uniform sampling from the replay buffer does not yield a uniform distribution over the overall state-action space since the sample distribution is usually non-uniform. Hence, it can be bad for TD error minimization and prioritized experience replay gives priority for samples based on TD error. Hence, regardless of that, the policy should select samples as uniformly as possible over the overall state-action space so that it can be used for convergence of Q.

---

> > ### Author Response · Authors · 2020-11-11
> > **Reply for reviewer 3 - Continue**
> >
> > •	If DAC is for improving exploration, ... on exploration methods in RL.
> >
> > Please note that the scope of exploration is very large, and our related work contains the entropy regularization and the diversity gain, and both categories fall under the exploration method.
> > Please also note that SAC itself can be considered a method enhancing exploration by adding the policy entropy to the net reward.  Also, DAC is a generalization of SAC since DAC reduces to SAC if $\alpha=1$ from (3). Hence, comparing with vanilla SAC is meaningful. However, as indicated by the reviewer, efforts were made to improve SAC’s performance by enhancing exploration further and SAC-Div [R3-4] is such an effort. We compared our algorithm with SAC-Div too in the paper.  SAC and SAC-Div are policy-based exploration enhancing methods. So, we compared our algorithm with RND, which is the state-of-the-art exploration method based on prediction error.  We will consider placing the RND result in the main paper if space allows.
> >
> > •	You define β in Equation 1 in $(0,\infty)$, can it really be infinite?
> >
> > $\beta \in (0,\infty)$ means that beta is larger than 0 and smaller than $\infty$ (finite $\beta$).
> >
> > •	In section 4.1. you define the target distribution ... What are the assumptions for this?
> >
> > Suppose that the current time is $t$. $q(a|s)$ is the sample action distribution computed from the current replay buffer at time $t$, and $\pi(a|s)$ is the current policy action distribution at time $t$. If we take action from $\pi(a|s)$ and store this sample $(s,a)$ in the replay buffer, then at time $t+1$, the sample action distribution $q(a|s)$ computed from the replay buffer will be a mixture of $q(a|s)$ at time $t$ and $\pi(a|s)$ at time $t$.  So, by defining the target distribution as the mixture of current $q(a|s)$ and current $\pi(a|s)$ and making the mixture as a uniform distribution, we can make the sample distribution in the replay buffer uniform over the entire state-action space as time goes. This means that we visit the entire state-action space as time goes.
> >
> > •	You prove the convergence ..., what are the assumptions for which the convergence is guaranteed?
> >
> > The conditions that are mentioned in the paper are as follows. They are typical. As usual Q-learning, every state-action pairs must be visited infinitely often, the learning rate should be decayed properly, and the reward function and the target entropy should be bounded.
> >
> > •	Why do you use sparse/delayed MuJoCo benchmarks, but not the original ones?
> >
> > We consider a pure exploration task of 4-room maze and sparse-rewarded tasks in order to show the exploration performance of DAC well. We choose sparse/delayed MuJoCo since it is often considered in previous works as cited in the paper.
> >
> > •	The variance across different seeds ..., why do you think it happens?
> >
> > We consider sparse reward setup, so the performance largely depends on the stochasticity of the policy. For sparse Mujoco examples, if there is a seed that successfully crosses the threshold and receives rewards early, then the agent will learn the policy faster to get rewards from nonzero reward samples. Then, the variance seems to be high until other seeds get rewards. For the pure exploration task of 4-room maze, there are 4 bottlenecks in the map. As in the case of the sparse Mujoco tasks, if the agent passes through a bottleneck early, then it will visit many new states. Thus, the variance in this case also seems to be high until other seeds passes through the bottleneck.
> >
> > •	For the adaptive α case, ... behind the left boundary?
> >
> > If the $\alpha$ goes close to 1, the ratio function also goes to 1 from equation (5) and it gets saturated for some tasks. Thus, we set an upper bound for $\alpha$ as 0.99. On the other hand, if the $\alpha$ goes close to 0, the policy entropy goes to 0 from equation (4). We want to maintain a proper amount of the policy entropy for exploration, so we set a lower bound for $\alpha$ as 0.5 in order to enforce the amount of the policy entropy, as explained in the paper.
> >
> > •	I think your paper can find an important application in Imitation Learning or off-line RL. ... which do something similar in those subfields?
> >
> > Thank you for the comment. I am aware of GAIL or other imitation learning/batch RL, but it is hard to find similar works that use the entropy of the mixture distribution. It would be great if the reviewer provides some examples for further study.
> >
> > [R3-1] Watkins, Christopher JCH, and Peter Dayan. "Q-learning." Machine learning 8.3-4 (1992): 279-292.
> >
> > [R3-2] Ahmed, Zafarali, et al. "Understanding the impact of entropy on policy optimization." ICML. 2019.
> >
> > [R3-3] Haarnoja, Tuomas, et al. "Soft Actor-Critic: Off-Policy Maximum Entropy Deep Reinforcement Learning with a Stochastic Actor." ICML. 2018.
> >
> > [R3-4] Hong, Zhang-Wei, et al. "Diversity-driven exploration strategy for deep reinforcement learning." NIPS. 2018.

---

> > > ### Comment · AnonReviewer3 · 2020-11-13
> > > **Response**
> > >
> > > Thank you for your clarifications!
> > >
> > > If I understood you correctly, you are trying to solve the exploration issue exploiting entropy regularisation in a novel manner.
> > >
> > > In this case, I would like to see a better positioning in the Related Work section not only mentioning work on the topic but relating your research to it and explaining pros/cons of the previous approaches.
> > >
> > > I still think that the paper does not provide enough experimental evidence to
> > > support the fact that the proposed approach helps with the exploration.
> > > While I agree, that we can see some improvement in Figure 1b, the rest of the experimental section simply compares the performance of the proposed method with SAC. But does DAC outperforms SAC because of enhanced exploration properties? Is it because of something else? We do not know. Also, are the MuJoCo tasks in question hard because of the exploration?
> > >
> > > Regarding comparison with RND, do you have any pointers to the literature which uses RND in a continuous state and action space case? The fact that RND works worse than SAC is a bit worrisome.
> > >
> > >
> > > *It would be great if the reviewer provides some examples for further study.*
> > > I do not know of any of the latest literature, but have a look at "Maximum entropy inverse reinforcement learning" by Ziebart et al.

---

> > > > ### Author Response · Authors · 2020-11-15
> > > > **Reply for reviewer 3**
> > > >
> > > > Thank you very much for understanding the motivation of our proposed method and additional feedback. Here is our reply:
> > > >
> > > > •	In this case, I would like to see a better positioning in the Related Work ... explaining pros/cons of the previous approaches.
> > > >
> > > > We will add a clear explanation of the motivation of the proposed method, the pros/cons of the simple entropy maximization in the related work section, as explained in the first comment in order to enhance the clarity of the paper.
> > > >
> > > > •	I still think that the paper does not provide enough experimental evidence. ...
> > > >
> > > > Please note that the final goal of RL is to achieve high scores for given tasks. For this, exploration techniques are needed to ensure that the policy does not converge to local optima, as explained in the first comment [R3-2]. Hence, we first showed the improvement of the exploration performance in a pure exploration task (4-room maze), and other experiments show that DAC has better return performance than SAC baselines (and other state-of-the art algorithms - please see Appendix F.3 too) on sparse-rewarded tasks. Please note that having  high scores on the sparse-reward tasks means that the policy can get rewards well without falling into local optima, which implies that the agent successfully explores more state-action pairs that have positive rewards. Therefore, we believe that the experiments considered in this paper fit well to the motivation. Please note that getting high scores in sparse-rewarded tasks has been widely used as a verification method of exploration in many previous exploration studies [R3-4, R3-5, R3-6, R3-7]. Also, in order to rule out the influence of factors other than exploration, we use the common simulation setup for DAC and SAC baselines except for the parts about entropy.
> > > >
> > > >
> > > > •	Regarding comparison with RND, ... The fact that RND works worse than SAC is a bit worrisome.
> > > >
> > > > We could not find an specific example in which RND was used in a continuous state-action space, but its variant, Random Expert Distillation (RED) [R3-8], was used for Imitation Learning for tasks with a continuous state-action space. Also, in the RND paper, we could not find some  assumptions or reasons by which RND can be used only in discrete state-action spaces [R3-6]. However, whereas the states of Atari have dramatic changes (room change or stage clear), the states of the continuous maze do not have such dramatic changes. RND explores rare states based on the prediction error for given state, so its exploration performance may not be significant if the states do not change dramatically. This is also the reason we set the main baselines as policy-based exploration techniques, SAC [R3-3] and SAC-Div [R3-4].
> > > >
> > > > •	It would be great if the reviewer provides some examples for further study. ...
> > > >
> > > > For further study, we consider a generalization of our method in order to deal with the entropy of the state-action distribution. Currently, many recent papers only consider one of the entropy of state distribution $d^\pi(s)$ or that of action distribution $\pi(a|s)$ only since they have much different properties (e.g. the state-based entropy is non-convex on $\pi$ and the action-based entropy is convex on $\pi$). However, both entropies can be handled simultaneously as one fused entropy that deals with the entropy of the state-action distribution, factorized as $\log d^\pi(s,a) = \log d^\pi(s) + \log \pi(a|s)$. Then, the generalization of our method for the fused entropy may be able to further enhance the exploration performance by considering the exploration on the entire state-action space.}
> > > >
> > > > Thank you again for your effort and time, and please see the contribution of our method. Basically, for sample-aware entropy maximization, we defined the sample distribution $q$ in equation (2) in the paper to formulate the mixture distribution and the problem. However, we did not actually compute $q$ from the sample buffer by using a method such as discretization and counting, which is not easy since the samples are continuous. Even if $q$ is obtained by counting, it is difficult to generalize such $q$ to yield the probability density value for arbitrary actions for any given state. Please note that we have to generalize $q$ as such because we need to compute $q$ of any policy action for any  given state. We circumvented this difficulty by exploiting  the ratio function $R^{\pi,\alpha}$ properly. Please note that this kind of approach is new as far as we know, and this approach significantly outperforms.
> > > >
> > > > [R3-5] Mazoure, Bogdan, et al. "Leveraging exploration in off-policy algorithms via normalizing flows." CoRL. 2020.
> > > >
> > > > [R3-6] Burda, Yuri, et al. "Exploration by random network distillation." ICLR. 2018.
> > > >
> > > > [R3-7] Ecoffet, Adrien, et al. "Go-explore: a new approach for hard-exploration problems." arXiv:1901.10995, 2019.
> > > >
> > > > [R3-8] Wang, Ruohan, et al. "Random Expert Distillation: Imitation Learning via Expert Policy Support Estimation." ICML. 2019.

---

> > > > > ### Comment · AnonReviewer3 · 2020-11-18
> > > > > **Response to the response**
> > > > >
> > > > >
> > > > > Thanks for incorporating the feedback into the revision. Please, find my comments below.
> > > > >
> > > > > - Policy ... which selects an action $a_t$ for a given state $s_t$...
> > > > >     - Why does the policy map to $(0, \infty)$?
> > > > > - In order to rule out the influence of factors other than exploration, we use the common simulation setup for DAC and SAC baselines except for the parts about entropy or divergence.
> > > > >     - I do not think that this claim is correct, and this is my main issue with the paper. We do not know if the gains come from improved exploration or not. Maybe they do, but this has to be shown in a proper scientific way devising a hypothesis and trying to falsify it. I believe, analysing the replay buffer, similarly to analysing the occupancy of the grid from Figure 1, is the first step towards this direction.
> > > > > - Note that having high scores on the sparse-rewarded tasks means that the policy can get rewards well without falling into local optima, which implies that the agent successfully explores more state-action pairs that have positive (or diverse) rewards.
> > > > >     - I understand the intuition. However, I think that this is not the only explanation. Moreover, assuming the claim is true, why don't we see the desired effect in SparseHopper/SparseAnt? Also, in one of the comments above, you refer to [R3-2] saying "Therefore, exploration for visiting different state-action pairs is important for RL, and the original entropy regularization encourages exploration". Can you, please, point me to the specific place in this paper which you had in mind?

---

> > > > > > ### Author Response · Authors · 2020-11-19
> > > > > > **Reply for reviewer 3**
> > > > > >
> > > > > > Thank you again for your valuable feedback.
> > > > > >
> > > > > > •	Why does the policy map to (0,∞)?
> > > > > >
> > > > > > The policy is a probability distribution over the action space for each given state, so it is a non-negative real-valued function as described in
> > > > > > \url{https://en.wikipedia.org/wiki/Probability_distribution#Continuous_probability_distribution}
> > > > > >
> > > > > > •	I do not think that this claim is correct, and this is my main issue with the paper. We do not know if the gains come from improved exploration or not. Maybe they do, but this has to be shown in a proper scientific way ....
> > > > > >
> > > > > > We provided the averaged learning curve of JS divergence curve $D_{JS}^\alpha (\pi||q)$ in Fig. 3 in order to empirically show that the proposed algorithm chooses more diverse action than the SAC baselines. Please note that DAC has larger JS divergence $D_{JS}^\alpha (\pi||q)$ than SAC in the overall learning time, and it means that the policy chooses actions from a distribution much different from the sample distribution in the replay buffer. But, as reviewer 3 said, in order to compare the exploration performance of DAC and the SAC baselines, we added the mean number of discretized state visitation curve in Fig. F.1. in Appendix F.1 (Please see the revised version).  For discretiziation, we simply consider the position of each Mujoco task and we can see from the Fig. F.1 that DAC visit more state blocks than the SAC baselines.
> > > > > >
> > > > > > Now, synthesizing the results of Fig. 3 and Fig. F.1, we can clearly conclude that the policy of DAC choose more diverse actions from the distribution far away from the sample action distribution $q$ as seen in Fig. 3, then DAC visits more diverse states than the SAC baselines as seen in Fig. F.1. Thus, DAC encourages better exploration and it yields better performance, which shows the superiority of the proposed sample-aware entropy regularization.
> > > > > >
> > > > > > •	I understand the intuition. However, I think that this is not the only explanation. Moreover, assuming the claim is true, why don't we see the desired effect in SparseHopper/SparseAnt? Also, in one of the comments above, you refer to [R3-2] saying "Therefore, exploration for visiting different state-action pairs is important for RL, and the original entropy regularization encourages exploration". Can you, please, point me to the specific place in this paper which you had in mind?
> > > > > >
> > > > > > For SparseMujoco tasks, the agent cannot get the reward sum higher than $1000$ points because the environment gives a maximum of 1 point if the exceeds the $x$-axis threshold. Thus, in the case of the SparseHopper task, SAC baselines also got almost the maximum score in most seeds. For SparseAnt task, we think there is a noticeable performance difference after 1.5M time-steps, as seen in Fig. 3(d) although the difference is small as compared to other environments. Also, please note that there is a clear difference in performance between DAC and SAC baselines on all the other considered tasks including DelayedMujoco tasks. We think that the results sufficiently show the policy of the proposed method falls into the local optima less than the SAC baselines.
> > > > > >
> > > > > > The authors of [R3-2] analyzed the effect of entropy on exploration  from a new perspective, as stated in the abstract of [R3-2]. They show that the policy can fall into local optima even with the exact gradient of the objective function  in Fig. 4 in Section 3.1 of [R3-2], depending on the geometry of the policy objective. Then, with their example they show that  the use of entropy regularization (i.e., imposing stochasticity on the policy) can make the objective function smoother and the entropy helps find better solution in Section 3.1 of [R3-2].
> > > > > >
> > > > > > Also, we can easily see that the entropy regularization encourages exploration. Let us consider again the  $1$-step MDP example, considered in the previous reply.
> > > > > > $s_0$ is the unique initial state, there are $N_a$ actions ($\mathcal{A} = \{A_0,\cdots,A_{N_a-1}\}$), $s_1$ is the terminal state, and $r$ is a deterministic reward function. We assumed that $N_a-1$ state-action samples are stored in the replay buffer as $\mathbf{R}=\{(s_0,A_0,r(s_0,A_0)),\cdots,(s_0,A_{N_a-2},r(s_0,A_{N_a-2}))\}$.
> > > > > > If we consider a deterministic policy that maximizes return, then it will never choose the last action $A_{N_a-1}$  since it always choose the action that has the best reward among the stored samples in the buffer. If the last action has the maximum reward, i.e. $r(s_0,A_{N_a-1}) > r(s_0,a)$ for all $a\neq A_{N_a-1}$ and $a\in\mathcal{A}$, then the policy never can get the maximum reward (i.e., the policy achieves local optima). Here, the entropy regularization imposes the stochasticity on the policy by making the policy choose actions that has lower probability more frequently since it adds $-\log\pi$ at the reward term. It clearly helps  choose the last action $A_{N_a-1}$, which is needed to escape the local optima by encouraging exploration.

---

### Official Review · AnonReviewer4 · 2020-10-28
**Efficient exploration for offline policy learning**

**Rating:** 6
**Confidence:** 3

**Review:**

Summary

This paper proposes a novel exploration method in off-policy learning. Compared to previous methods which do not take care into account the distribution of the samples in the replay buffer, the proposed method maximizes the entropy of the mixture of the policy distribution and the distribution of the samples in the replay buffer, hereby making exploration efficient.

Reasons for score

I vote for accepting the paper. The paper proposes an intuitive and efficient exploration method that generalizes existing methods, including them as special cases. The authors provide a theoretical guarantee (Theorem 1) that the policy obtained from the iteration of evaluation and improvement under this new regime converges to the optimal policy.  The presentation is clear and concrete, and the experiments are convincing.

Pros

The experiment results are not limited to just showing that the proposed method achieves higher reward than state of the art methods, but they also address important questions such as
(i) the pure exploration when rewards are assumed to be 0
(i) the necessity of the adaptation of alpha, the parameter that controls the ratio of the current policy to the sample distribution in the target distribution.
(ii) the effect of controlling alpha, the entropy weighting factor beta, and the control coefficient c (required for adapting alpha), and also, the robustness of the proposed method to these parameters.

The authors have stated the experiment details clearly and the results are convincing.

Cons

The methodology part in Section 3 and 4 could be improved. Some notations are confusing.
(a) In Section 3, the policy \pi is defined as a function from S to A. It looks like it is a  fixed function over time.
(b) An explanation on the definition of J_{pi 1}(pi 2) would be helpful,e.g.,  J_{pi 1}(pi 2) is value of J(pi_2) computed under pi_1.

Minor Comments

It would be good to add the line of SAC and SAC-Div in Figure 5 (c ) to show that the performance of DAC with adaptive alpha is robust to control coefficient c. For now, one has to go back to Figure 4 (b) to check that most of the case (when c is not 0), DAC with adaptive alpha performs better than SAC and SAC-Div.

In Section 6 in the 5th line, J(\pi) should be specified as “J(\pi) in (1)”. It is done in the next sentence, but I prefer that it is done when it first appears. It was confusing

---

> ### Author Response · Authors · 2020-11-11
> **Reply for reviewer 4**
>
> Thank you for the positive feedback and for highlighting the strength of our proposed method. Confusing notation and figure will be fixed as soon as possible.

---

### Official Review · AnonReviewer2 · 2020-10-29
**Diversity Actor-Critic: Sample-Aware Entropy Regularization for Sample-Efficient Exploration**

**Rating:** 5
**Confidence:** 4

**Review:**

This paper proposes diversity actor-critic (DAC) for exploration in reinforcement learning. The main idea of the proposed algorithm is to take advantage of the previous sample distribution from the replay buffer for sample-efficient exploration. The authors provide convergence analysis of DAC and conduct empirical investigations on several benchmarks.

Pros

The idea of using previous sample distribution from the replay buffer for better exploration seems interesting. The proposed exploration bonus $\mathcal{H}(q^{\pi, \alpha}_{\text{target}})$ can be decomposed into three terms as shown in (4). Since the last term does not depend on $\pi$, intuitively this exploration bonus encourages the exploration of $\pi$ (first term), and tries to make $\pi$ different with previous policies approximated by the replay buffer (second term). The authors provide a reasonable method to optimized the proposed objective, which can be naturally combined with state-of-the-art algorithms like SAC.

Cons

1. Theorem 1 seems misleading. The diverse policy iteration can only guarantee the converge to the optimal policy with respect to the regularized value function, not the optimal policy of the original problem. The authors should make the definition of $\pi^*$ clear.

2. It’s hard to see the motivation of using a mixture of $q$ and $\pi$. Could you explain more about this choice?

3. It’s worth to provide the results of SAC-div with JS divergence as it’s more similar to the proposed objective (4).

4. The experiment results are not convincing enough as some important baselines are missing. For example, [1] also uses a mixture of previous polices to encourage exploration with strong theoretical guarantees. I believe this is closely related to the proposed algorithms.
Also, the experiment results are not very promising compared with the baseline algorithms based on SAC.

[1] Hazan, E., Kakade, S., Singh, K. and Van Soest, A., 2019, May. Provably efficient maximum entropy exploration. In International Conference on Machine Learning (pp. 2681-2691).

Other suggestions

The main idea of the proposed method is to make the current policy different with previous policies. The paper uses a nonparametric method  (2) to approximate the previous policies. I think it’s also worth to try parametric $q$. For example, $q$ could be learned by fitting the replay buffer, or use a moving average of previous policies.

---

> ### Author Response · Authors · 2020-11-11
> **Reply for reviewer 2**
>
> Thank you for the valuable comments and constructive feedbacks. We provide our feedback below:
>
> •	Theorem 1 seems misleading. The diverse policy iteration can only guarantee the converge to the optimal policy with respect to the regularized value function, not the optimal policy of the original problem. The authors should make the definition of $\pi^*$ clear.
>
>    We define the objective function as the sample-aware entropy regularized return as (3) (not just return), so the optimal policy $\pi^*$ in this paper maximizes (3) as explained Theorem 1.  Please note that the optimal policy of SAC is also defined as the policy that maximizes the entropy regularized return (1) [R2-1]. In order to guarantee convergence to optimal policy for return maximization, we have to shrink the entropy coefficient $\beta$ to $0$.
>
> •	It’s hard to see the motivation of using a mixture of q and π. Could you explain more about this choice?
>
>    The detailed motivation of using the mixture distribution is as follows.  In order to guarantee the convergence of Q-learning [R2-2], there is a key assumption: Each state-action pair must be visited infinitely often. If the policy does not visit diverse state-action pairs many times, it converges to local optima. Therefore, exploration that promotes visiting different state-action pairs is important for RL, so it is better to draw new samples while avoiding samples that have been drawn many times before.
>
>    The simple policy entropy maximization will choose all actions with equal probability without considering the previous action samples. In contrast, if we consider maximizing the entropy of the mixture of $\pi$ and $q$ (it becomes the future sample distribution since the buffer stores samples generated by $\pi$), $\pi$ should choose actions rare in the buffer with high probability and actions stored many times in the buffer with low probability to make the mixture distribution uniform. Hence, it considers the samples already stored in the buffer in choosing current action and encourages sample-efficient exploration. We provide a simple example below:
>
>    Let us consider a simple 1-step MDP in which $s_0$ is the unique initial state, there exist $N$ actions ($a_0\in(\{1,\cdots,N\})$), $s_1$ is the terminal state, and $r$ is the deterministic reward function. Then, there exist $N$ state-action pairs in total and let us assume that we already have $N-1$ state-action samples in the replay buffer as $R=(\{(s_0,1,r(s_0,1)),\cdots,(s_0,N-1,r(s_0,N-1))\})$. In order to estimate the Q-function for all state-action pairs, the policy should sample Action $N$ (After then, we can reuse all samples infinitely to estimate Q). Here, we will compare two exploration methods.
>
> 1)	First, if we consider the simple entropy maximization, the policy will choose all actions with equal probability $1/N$ (uniformly) since $\max_\pi \mathcal{H}(\pi) =\min_\pi KLD(\pi||U)$ is achieved when $\pi=U$, where $U$ is a uniform distribution. Then, $N$ samples should be taken on average by the policy to visit Action $N$.
>
> 2)	Consider the sample-aware entropy maximization as in our paper. Here, the sample action distribution $q$ in the buffer is defined as $q(a_0|s_0)=1/(N-1)$ for $a_0\in\{0,\cdots,N-1\}$, and $q(N|s_0)=0$.
>
>    Now, we set the target sample distribution as the mixture of $\pi$ and $q$, as $q_{target}^{\pi,\alpha}=\alpha\pi+(1-\alpha) q$, and we set $\alpha=1/N$. Then, in order to maximize the entropy of the target sample distribution $\max_\pi \mathcal{H}( q_{target}^{\pi,\alpha} )  =  \min_\pi KLD( q_{target}^{\pi,\alpha} ||U)$ , the policy distribution should be $\pi(N|s_0)=1$ to make $q_{target}^{\pi,\alpha}$ uniform. Thus, it only needs one sample to visit Action $N$.
>
>    In this way, the simple entropy regularization is sample-inefficient for off-policy RL, and the proposed sample-aware entropy regularization enhances the sample-efficiency for exploration by using the previous sample distribution and choosing proper $\alpha$. With this motivation, we propose the sample-aware entropy regularization and the corresponding $\alpha$-adaptation. Therefore, the method in this paper addresses sample-efficient exploration.

---

> > ### Author Response · Authors · 2020-11-11
> > **Reply for reviewer 2 - Continue**
> >
> > •	It’s worth to provide the results of SAC-div with JS divergence as it’s more similar to the proposed objective (4).
> >
> > In order to compute the JS-divergence, the ratio function $R^\alpha$ is necessary, and computing the JS-divergence based on $R^\alpha$ practically is one of the main contributions of our paper. Theorem 2 is about this contribution. If we want to use JS divergence for SAC-div, this should use our technique proposed in this paper.  The algorithms in [R2-3] use the KL divergence or simple MSE for their simulation because the computation of the KL divergence or MSE can be done without special technique.  Thus, our baseline just followed the algorithm as proposed in [R2-3].  Anyway, we agree on that the comparison suggested by the reviewers is helpful for better ablation study. We will add the result of SAC-Div with JS-divergence as soon as possible.
> >
> > •	The experiment results are not convincing enough as some important baselines are missing. For example, MaxEnt also uses a mixture of previous polices to encourage exploration with strong theoretical guarantees. I believe this is closely related to the proposed algorithms.
> >
> > Please note that there exist so many algorithms developed for better exploration, and we can divide them into two groups: 1) state-based exploration and 2) action(policy)-based exploration. Max-Ent [R2-4] is a state-based exploration and our method is a action-based exploration. MaxEnt estimates and regularizes the entropy of count-based mixture state distribution $d^{\pi_{mix}}(s)$. On the other hand, our method mainly concerns the improvement of the existing entropy regularization of the policy distribution $\pi(a|s)$ by using the mixture action distribution $q_{target}^{\pi,\alpha}=\alpha\pi+(1-\alpha) q$. Then, the entropy of state mixture distribution and the entropy of action mixture distribution are much different since the action entropy is convex on $\pi$ but the entropy of state distribution is non-convex on $\pi$ [R2-4]. In addition, both methods can be applied separately since the state-action distribution can be factorized as $\log d^\pi(s,a) = \log d^\pi(s) + \log \pi(a|s)$ as MaxEnt uses SAC for its planning oracle.
> >
> > For these reasons, we compared our method mainly to action-based exploration methods: SAC [R2-1] that considers the original policy entropy regularization (DAC is a generalization of SAC) and SAC-Div [R2-3] that diversifies the policy action distribution from the buffer that are closely related to our proposed method. Also, if it's just to compare exploration performance, we already provided the comparison to another state-of-the-art state-based exploration method, random network distillation (RND) [R2-5] in Appendix. The result shows that our method is much superior to RND.
> >
> > •	The experiment results are not very promising compared with the baseline algorithms based on SAC.
> >
> > We compared the performance for pure exploration tasks and sparse-rewarded tasks. As shown in the paper, the proposed method DAC significantly outperforms SAC baselines for most tasks: DAC explores more states that SAC cannot reach in the continuous maze task. Especially, SAC even fails to learn SparseHalfCheetah for some seeds and converges too early in HumanoidStandup and DelayedMujoco tasks, but DAC learns well for those tasks with drastic performance gap. Therefore, we believe that the experimental results in the paper already show the superiority of the proposed method as compared to the previous methods.
> >
> > [R2-1] Haarnoja, Tuomas, et al. "Soft Actor-Critic: Off-Policy Maximum Entropy Deep Reinforcement Learning with a Stochastic Actor." ICML. 2018.
> >
> > [R2-2] Watkins, Christopher JCH, and Peter Dayan. "Q-learning." Machine learning 8.3-4 (1992): 279-292.
> >
> > [R2-3] Hong, Zhang-Wei, et al. "Diversity-driven exploration strategy for deep reinforcement learning." NIPS. 2018.
> >
> > [R2-4] Hazan, Elad, et al. "Provably efficient maximum entropy exploration." 36th International Conference on Machine Learning, ICML 2019.
> >
> > [R2-5] Burda, Yuri, et al. "Exploration by random network distillation." ICLR. 2018.

---

> > > ### Comment · AnonReviewer2 · 2020-11-21
> > > **Response**
> > >
> > > Thank you for your clarifications. But I still have some concerns.
> > >
> > > For the pure exploration problem, why not MaxEnt is a proper baseline? I am not fully convinced that this baseline should be ignored just because it applies a different technique with DAC. I do think MaxEnt and DAC share similar intuition, but it seems to me that MaxEnt tackle the pure exploration problem in a more elegant way.
> > >
> > > Furthermore, I am still wondering why using a mixture of $q$ and $\pi$ in the regularization. If we ignore the term that does not effect $\pi$, the regularization (4) basically says 1. encourage exploration of $\pi$ by maximizing the entropy, and 2. make $\pi$ away from previously explored actions. If I want to design a regularization implementing such intuition, why it must use a mixture of $q$ and $\pi$, which brings extra difficulty in the optimization problem? Why not just fitting a policy using the replay buffer and regularize $\pi$ away from that? I am not saying a mixture of $q$ and $\pi$ is bad, but I think the authors should provide some evidence (maybe just simple experiment results) to show the effectiveness of the current algorithm choice.
> > >
> > > Finally, I think the example you made is not very convincing. The replay buffer, which is used to get $q$, is also collected by the agent's policy $\pi$. So if you just consider $\pi$ that maximizes the entropy, it will also select all actions in $N_a$ steps in expectation.

---

> > > > ### Author Response · Authors · 2020-11-21
> > > > **Reply for reviewer2**
> > > >
> > > > Thank you for your valuable feedback.
> > > >
> > > > •	For the pure exploration problem, why not MaxEnt is a proper baseline? I am not fully convinced that this baseline should be ignored just because it applies a different technique with DAC. I do think MaxEnt and DAC share similar intuition, but it seems to me that MaxEnt tackle the pure exploration problem in a more elegant way.
> > > >
> > > > As reviewer 2 suggest, we added the pure exploration comparison of DAC and MaxEnt on the continuous 4-room maze task. MaxEnt considers maximizing the entropy of state mixture distribution $d^{\pi^{mix}}$ by setting the reward functional in [R2-4] as $-\log d^{\pi_{mix}}(s) + c_M$, where $c_M$ is a smoothing constant. For MaxEnt, we compute the reward functional at each iteration by using Kernel density estimation with a bandwidth $0.1$ as stated in [R2-4] on previous $10000$ states stored in the buffer, and we use $c_M=0.01$. Fig. F.1. shows the number of state visitation on the continuous 4-room maze task. Here, please note that MaxEnt visits slightly more states than RND, but DAC can visit much more states than MaxEnt, and it shows the superiority of DAC. We also think that MaxEnt handles the entropy of state mixture distribution well, but the entropy domains of the two algorithms are different (DAC considers the mixture distribution of action distribution and MaxEnt considers the mixture distribution of state distribution.) and their properties are also different as we said. Also, the result of the pure exploration comparison shows that DAC visited the largest number of states, so we think that our work is sufficiently worthy in terms of exploration.
> > > >
> > > > •	Furthermore, I am still wondering why using a mixture of $q$ and $\pi$ in the regularization. If we ignore the term that does not effect $\pi$, the regularization (4) basically says 1. encourage exploration of $\pi$ by maximizing the entropy, and 2. make $\pi$ away from previously explored actions. If I want to design a regularization implementing such intuition, why it must use a mixture of $q$ and $\pi$, which brings extra difficulty in the optimization problem? Why not just fitting a policy using the replay buffer and regularize $\pi$ away from that? I am not saying a mixture of $q$ and $\pi$ is bad, but I think the authors should provide some evidence (maybe just simple experiment results) to show the effectiveness of the current algorithm choice.
> > > >
> > > > Please note that SAC-Div [R2-3] is the algorithm that learns  task by considering exactly in the same way as reviewer 2 said. It simply learns the policy $\pi$ based on a baseline algorithm (SAC in this paper) and add a divergence regularization term in order to choose action samples from the distribution away from the sample distribution in the replay buffer. Thus, we consider SAC-Div as one of the baselines and we numerically show that our proposed method is superior to a simple divergence regularization on various tasks. Also, we consider the performance comparison of DAC and SAC-Div with both KL/JS divergences in Fig. 5(d), and the result show that DAC has a higher performance increase than SAC-Div with both divergences. Thus, we think that it sufficiently shows what reviewer 2 said.
> > > >
> > > > •	Finally, I think the example you made is not very convincing. The replay buffer, which is used to get $q$, is also collected by the agent's policy $\pi$. So if you just consider $\pi$ that maximizes the entropy, it will also select all actions in $N_a$ steps in expectation.
> > > >
> > > > Please note that our example is a simple case in order to show that the sample-aware entropy has the superiority as compared with the policy entropy, when the sample distribution in the buffer is non-uniform. Thus, we consider the case that non-uniform $q$ is given. Furthermore, even if we consider the case that we collects samples from the beginning without given $q$, the probability of choosing all actions in $N_a$ steps at once by using $\pi$ that maximizes the entropy (the uniform distribution) is   $\frac{(N_a-1)!}{N_a ^{N_a-1}}$ only. Thus, it is hard to select all actions in $N_a$ steps, and the expectation of the number of samples to choose all actions is strictly larger than $N_a$. In contrast, the sample aware entropy maximization enables to choose all actions in $N_a$ steps if we repeatedly choose one sample from $\pi$ and set $\alpha=(N_a-t)/N_a$ at each step $t\geq 1$. It is because the sample aware entropy maximization with such $\alpha$ always makes $\pi(a|s_0)=0$ if $a$ exists in the replay buffer at each step. Thus, the sample-aware entropy  enhances the sample-efficiency in this case.

---

### Official Review · AnonReviewer1 · 2020-11-11
**Novel idea with some clarity and technical issues**

**Rating:** 5
**Confidence:** 4

**Review:**

This paper considers the exploration efficiency issues in off-policy deep reinforcement learning (DRL). The authors identify a sample efficiency limitation in the classical entropy regularization, which does not take into account the existing samples in the replay buffer. To avoid repeated sampling of previously seen scenarios/actions, the authors propose to replace the current policy in the entropy term with a mixture of the empirical policy estimation from the replay buffer and the current policy, and term this approach as sample-aware entropy regularization. The authors then propose a theoretical algorithm called sample-aware entropy regularized policy iteration, which is a generalization of the soft policy iteration (SPI) algorithm, and show that it converges assuming that the empirical policy estimation is fixed. A practical algorithm based on the sample-aware entropy regularized policy iteration, called Diversity Actor-Critic (DAC), is then proposed. This algorithm is a generalization of the well-known soft actor-critic (SAC) algorithm. Finally, numerical experiments show that DAC outperforms SAC and other SOTA RL algorithms, and some ablation studies are also provided to demonstrate the effect of hyper-parameter choices in DAC.

In general, the approach is novel to my knowledge and the high level idea of using mixed policies in the entropy regularization to avoid repeated sampling and encourage unseen scenarios/actions is also interesting and reasonable. However, there are some clarity and technical issues that should be addressed and improved, as listed below:
1. The authors study finite horizon MDPs, for which the optimal policy should be non-stationary in general. However, the authors only consider stationary policies. Instead, the authors should either change the underlying setting to infinite horizon MDPs or consider non-stationary policies.
2. In (2), $s_t$ should be replaced by an arbitrary $s$ in the state space. Otherwise there may be contradicting definitions of the policy $q$ if $s_t$ and $s_{t’}$ are equal for some two different timestamps $t$ and $t’$. And in (3), it is better to write the $q_{\rm target}^{\pi,\alpha}$ in the entropy term as $q_{\rm target}^{\pi,\alpha}(\cdot|s_t)$, to be consistent with (1).
3. It’s not very clear why the authors propose to estimate $R^{\pi,\alpha}$ with some (neural network) parametrized $R^{\alpha}$. The authors mention that one can only estimate $R^{\pi_{\rm old},\alpha}$ for the previous policy $\pi_{\rm old}$ in practice. However, since in $R^{\pi,\alpha}$, all the quantities including $\pi$, $q$ and $\alpha$ are known, I’m confused why one cannot evaluate it directly. On a related point, it’s not very clear why the estimation procedure for $\eta$ (the parameter of $R^{\alpha}$) using hat $J_{R^{\alpha}}(\eta)$ makes sense. The form of hat $J_{R^{\alpha}}(\eta)$ looks like an entropy term extracted from the $J_{\pi_{\rm old}}$ function, but it’s unclear why maximizing it gives a good estimation of $R^{\pi,\alpha}$. Some more explanations are needed.
4. There seem to be several errors (at least inaccuracies) in the proof of Theorem 1 (in the Appendix). Firstly, in the proof of Lemma 1, the term “correctly estimates” is not very accurate, and should be simply stated as something like “equals”. Also, it’s not very clear when the assumption $R^{\alpha}\in(0,1)$ can be guaranteed (e.g., using Gaussian/soft-max policies?). Secondly, in the main proof of Theorem 1, convergence of $Q^{\pi_i}$ to some $Q^{\star}$ is correct, but this does not immediately imply convergence of $J_{\pi_i}$, let alone the convergence of $\pi_i$ to some policy $\pi^\star$. On a related point, the proof for the optimality of $\pi^\star$ in terms of $J$ is not clear. In particular, it is not clear why (7) and Lemma 2 implies the chained inequality $J_{\pi_{\rm new}}(\pi_{\rm new})\geq J_{\pi_{\rm old}}(\pi_{\rm new})\geq J_{\pi_{\rm old}}(\pi_{\rm old})$. I understand that the authors may feel that the proofs are similar to that of SPI, but indeed there are several significant differences (e.g., the definitions of $\pi_{\rm new}$ and $J_{\pi}$). More rigorous proofs are needed for these claims.
5. In Section 5, it is unclear why the authors need to include the parameter $c$, how to choose it and what it serves for. Some additional explanations are needed.
6. On a high level, the eventual goal of the paper is not clearly stated. From the experiments, it seems that the average episode reward is the actual goal of concern. However, the problem setting and the theoretical results (Theorem 1) seem to indicate that the problem of concern is the discounted entropy regularized reward. Some discussion about this is needed.

Finally, here are some more minor comments and suggestions:
1. In the analysis of the sample-aware entropy regularized policy iteration, the authors assume that $q$ is fixed. However, in practice, especially in the long run (as concerned in the analysis), such an assumption will not hold (even in just an approximate sense). Can you still obtain some sort of convergence when taking into account the $q$ changes?
2. Why do you need to divide the reward and entropy regularization term in $Q^{\pi}$ by $\beta$?
3. It’s better to write out the “binary entropy function $H$" explicitly for clarity.
4. At the beginning of Section 4.3, “propoed” should be “proposed”, and In Section 5, “a function $s_t$” should be “a function of $s_t$”.
5. Some high level explanations on why the $(1-\alpha)$ term can also be dropped in (8) will be helpful.
6. The theoretical results only show that the algorithm converges, which is already guaranteed by SPI. Is there any possibility to show that there is also some theoretical improvement?

So in short, the paper proposes an interesting modification of the max-entropy regularization framework, but contains several technical and clarity issues. Hence I think it is not yet ready for publication in its current form.

---

> ### Author Response · Authors · 2020-11-12
> **Reply for reviewer 1**
>
> Thank you for the valuable comments and constructive feedbacks. We provide our feedback below:
>
> •	The authors study finite horizon MDPs, ... change the underlying setting to infinite horizon MDPs or consider non-stationary policies.
>
> Our paper considers infinite horizon MDP as basic RL setup, and the proof of the diverse policy iteration also assumes an infinite horizon MDP. We will mention it in Section 3.1 as the reviewer says. Episode length $T$ also should be $\infty$ but many paper use “$T$” though they consider infinite horizon MDP and we followed this convention. In the revised we change $T$ to $\infty$.
>
> •	In (2), $s_t$ should be replaced by an arbitrary $s$ in the state space.
>
> We will change the both confusing notations.
>
> •	It’s not very clear why the authors propose to estimate $R^{\pi,\alpha}$ ... I’m confused why one cannot evaluate it directly.
>
> We defined the sample distribution $q$ in equation (2). However, we do not actually compute $q$ from the sample buffer by using a method such as discretization and counting for continuous samples, which is not simple. Even if $q$ is obtained by counting, it is difficult to generalize such $q$ to yield the probability density value for arbitrary actions for any given state. Please note that we have to generalize $q$ as such because we need to compute $q$ of any policy action for any  given state. We circumvented this difficulty by defining  the ratio function $R^{\pi,\alpha}$.
>
> Please note that what we need is the entropy of the target distribution, as seen in equation (3). We show that this mixture entropy can be decomposed as $\mathcal{H}(\pi) + D^\alpha_{JS}(\pi||q) + (1-\alpha) \mathcal{H}(q)$.  Then, we express $D^\alpha_{JS}(\pi||q)$ in terms of $R^{\pi,\alpha}$ not $q$, and $H(q)$ can be handled without computing $q$ explicitly again by using the property of the ratio function as eq (B.7) in Appendix. Hence, at least regarding the entropy of the target distribution, $R^{\pi, \alpha}$ is sufficient.
>
> Now, the objective function for policy update is given by equation (7), and note that $\pi$ in $R^{\pi,\alpha}$ is the optimization variable. Since we do not compute $q$, we can express $R^{\pi,\alpha}$ in terms of $\pi$ explicitly for optimization. We circumvented this difficulty by showing that maximizing the original objective function (7) is equivalent to maximizing the alternative objective function equation (8) in which $R^{\pi_{old},\alpha}$ appears. And, $R^{\pi_{old},\alpha}$ is estimated by using a neural network $R^\alpha$. Note that this neural network has generalization effect.
>
> •	On a related point, it’s not very clear why the estimation procedure for $\eta$ ... Some more explanations are needed.
>
> Please note that $J(R^\alpha)$ is just an $\alpha$-skewed JS divergence except some constant terms. In the $\alpha=0.5$ case, it becomes the usual JS divergence, which is considered in GAN [R1-1]. [R1-1] has shown that the ratio function for $\alpha=0.5$ can be estimated by maximizing the JS divergence. In a similar way to that in [R1-1], we can show that maximizing $J(R^\alpha)$ can estimate our ratio function as below:
>
> For given $s$, $J(R^\alpha(s,\cdot)) = \int_a \alpha \pi(a|s) \log R^{\alpha}(s,a) + (1-\alpha) q(a|s) \log (1-R^{\alpha}(s,a)) da$. The integrand is in the form of the function $y\rightarrow a \log y + b\log (1-y)$ with $a=\alpha\pi$ and $b=(1-\alpha)q$, and for any positive $(a,b)$, the function its maximum at $a/(a+b)$. Thus, the optimal ${R^{\alpha}}$$^*$ maximizing $J(R^\alpha (s,\cdot))$ is   ${R^{\alpha}}$$^*$ $ (s,a) = \alpha\pi / (\alpha\pi+(1-\alpha)q)$.
>
> •	There seem to be several errors in Theorem 1 ... More rigorous proofs are needed for these claims.
>
> 1)	The term “correctly estimates” will be changed as reviewer 1 said.
>
> 2)	$R^\alpha \in (0,1)$ is guaranteed when $\pi$ and $q$ are non-zero for all state-action pairs. For practical implementation, we clipped the ratio function as $(\epsilon,1-\epsilon)$ for small $\epsilon>0$ since some $q$ values can be close to zero before the replay buffer stores a sufficient amount of samples. $\pi$ is always non-zero since we consider Gaussian policy.
>
> 3)	Assume an arbitrary state $s\in\mathcal{S}$. From the policy update, $J_{\pi_{old}}(\pi_{new})\geq J_{\pi_{old}}(\pi_{old}$ as stated int the proof of Lemma 2. Next, from the eq. (7), all terms are the same for $J_{\pi_{new}}(\pi_{new}(\cdot|s))$ and $J_{\pi_{old}}(\pi_{new}(\cdot|s))$ except $\beta E_{a\sim \pi_{new}(\cdot|s)}[Q^{\pi_{new}}(s,a)]$ in $J_{\pi_{new}}(\pi_{new}(\cdot|s))$ and $\beta E_{a\sim \pi_{new}(\cdot|s)}[Q^{\pi_{old}}(s,a)]$ in $J_{\pi_{old}}(\pi_{new}(\cdot|s))$.
>
> Since $ Q^{\pi_{new}}(s,a) \geq Q^{\pi_{old}}(s,a)$ for any $(s,a) \in \mathcal{S}\times\mathcal{A}$ by Lemma 2, $ J_{\pi_{new}}(\pi_{new}(\cdot|s)) \geq J_{\pi_{old}}(\pi_{new}(\cdot|s))$. Thus, $J_{\pi_{new}}(\pi_{new}(\cdot|s)) \geq J_{\pi_{old}}(\pi_{new}(\cdot|s)) \geq J_{\pi_{old}}(\pi_{old}(\cdot|s))$ for any state $s\in\mathcal{S}$.

---

> > ### Author Response · Authors · 2020-11-12
> > **Reply for reviewer 1 - Continue**
> >
> > •	In Section 5, it is unclear why the authors need to include the parameter $c$, how to choose it and what it serves for. Some additional explanations are needed.
> >
> > We want to maintain the target entropy at a certain level to explore the state-action space well. In order to do so, we learn $\alpha$ to minimize $\mathcal{H}(q_{target}^{\pi,\alpha}) - \alpha c
> > $, and the role of $c$ is to maintain the target entropy $\mathcal{H}(q_{target}^{\pi,\alpha})$ close to $\alpha c$. Thus, $c$ is important to control the amount of the target entropy, and the ablation study for parameter $c$ is  given in Section 6.3.
> >
> > •	On a high level, the eventual goal of the paper is not clearly stated. ... Some discussion about this is needed.
> >
> > Usual RL aims to maximize the discounted return, but in order to guarantee the convergence of Q-learning [R1-2], there is a key assumption: Each state-action pair must be visited infinitely often. If the policy does not visit diverse state-action pairs many times, it converges to local optima. Therefore, exploration for visiting different state-action pairs is important for RL, and the original entropy regularization encourages exploration [R1-3]. We provided further detailed motivation our paper in the reply for reviewer 3. Thus, our paper concerns about the improvement of exploration, so we first tested our algorithm on pure exploration tasks to show that our proposed method explores the state-action space better than the simple entropy regularized objective. Then, we compared the performance on sparse-rewarded tasks, and having a high score on the sparse-reward tasks means that the policy can get rewards well without falling into local optima. Therefore, the experiments considered in this paper fit well to the motivation.
> >
> > •	In the analysis of the sample-aware entropy regularized policy iteration, the authors assume that $q$ is fixed. ... Can you still obtain some sort of convergence when taking into account the q changes?
> >
> > The purpose of assuming that q is fixed is slightly different from what reviewer 1 said. Our purpose for the assumption is that $q$ changes very slowly since the replay buffer has much large size ($\sim 1M$) than samples generated in one iteration or episode as mentioned in the paper, so we can think $q$ is approximately fixed in the near iteration when the policy is updated. If $q$ is changed in long run as the reviewer 1 said, the policy will go to the optimal policy of the objective function for changed $q$. However, we think that changing $q$ in long run is not a critical issue since we will make the entropy weighting coefficient $\beta$ zero if the policy explores the state-action space sufficiently to maximize the discounted return finally.
> >
> > •	Why do you need to divide the reward and entropy regularization term in $Q^\pi$ by $\beta$ ?
> >
> > We just follow the setup of SAC [R1-4]. SAC gives a reward scaling as $1/\beta$ instead of setting the entropy coefficient as $\beta$.
> >
> > •	The theoretical results only show that the algorithm converges, which is already guaranteed by SPI. Is there any possibility to show that there is also some theoretical improvement?
> >
> > Diverse policy iteration (DPI) is a generalization of SPI and DPI needs to deal with the ratio function $R^{\pi,\alpha}$. Please note that the proof of policy improvement is needed since it is hard to compute $Q^\pi$ for current policy $\pi$ (i.e., $\pi$ as optimization variable) (If we can compute the true $Q^\pi$, we can just maximize it). We have Theorem 2 to improve the original objective function (3) since we only have $R^{\pi_{old},\alpha}$. Thus, theorem 2 is essential to complete DPI and it is clearly distinct from SPI. Please see the reply to the third comment.
> >
> > “As seen, we circumvented several difficulties in incorporating the sample action distribution in the replay buffer and successfully implemented the sample-aware entropy regularization, which significantly improves performance. We will revise the paper to clarify the issues that the reviewer brought.  Thank you again for the detailed comments and efforts.”
> >
> > [R1-1] Goodfellow, Ian, et al. "Generative adversarial nets." NIPS. 2014.
> >
> > [R1-2] Watkins, Christopher JCH, and Peter Dayan. "Q-learning." Machine learning 8.3-4 (1992): 279-292.
> >
> > [R1-3] Ahmed, Zafarali, et al. "Understanding the impact of entropy on policy optimization." ICML. 2019.
> >
> > [R1-4] Haarnoja, Tuomas, et al. "Soft Actor-Critic: Off-Policy Maximum Entropy Deep Reinforcement Learning with a Stochastic Actor." ICML. 2018.

---

> > ### Comment · AnonReviewer1 · 2020-11-22
> > **Response to response (part 1/2)**
> >
> > Thanks a lot for the explanations and the updates in the draft! In particular, the explanations and revisions on $R^{\pi,\alpha}$, $R^{\alpha}$ and $\eta)$ are now much clearer than in the original draft, which is good. Also, I think the explanations on the chained inequality is valid. However, I'm still not fully convinced by the explanations on the technical proof. In particular, it's still not clear why the convergence $\pi_i$ to some (optimal) policy $\pi^\star$ is valid, as pointed out in my original review. And this is the key to all other arguments in the proof. Btw, FYI, it seems that eq. (7) should now be eq. (8) in the revised draft.

---

> > > ### Author Response · Authors · 2020-11-23
> > > **Reply for reviewer 1**
> > >
> > > Thank you for your valuable feedback again.
> > >
> > > We provided more mathematically complete proof for Theorem 1. Please see the revised version since the proof is too long to write it here. We now think that the revised version of the proof of Theorem 1 contains information enough in order to follow the proof.

---

### Author Response · Authors · 2020-11-17
**Common reply**

We thank all reviewers for their valuable comments and constructive feedbacks.

Based on the feedbacks of all reviewers, we revised the paper and uploaded the revised paper.

In the revised version, we enhanced the clarity of Theorem 1 in Appendix A and provided detailed implementation in Appendix B. Then, we clearly stated the motivation of this paper in Section 2/Section 4.1, and the purpose of using the ratio function in Section 4.3. In addition, we stated why we could consider the sparse-rewarded tasks as a verification method of exploration in Section 6.2, and we provided an additional ablation study on the effect of JS divergence in Section 6.3. Also, we added the learning curve that shows the mean number of discretized state visitation in order to see the exploration performance on sparse Mujoco tasks in Appendix F.1. Finally, we added possible further study in Section 7.

---

### Decision · Program_Chairs · 2021-01-07
**Final Decision**

**Decision:**

Reject

**Comment:**

First, I'd like to thank both the authors and the reviewers for extensive and constructive discussion. The paper proposes a generalization of SAC, which considers the entropy of both the current policy and the action samples in the replay pool. The method is motivated by better sample complexity, as it avoids retaking actions that already appear in the pool. The paper formulates a theoretical algorithm and proves its convergence, as well as a practical algorithm that is compared to SAC and SAC-Div in continuous sparse-reward tasks.

Generally, the reviewers found the method interesting. After rounds of discussion and revisions, the reviewers identified two remaining issues. Theoretical analysis still requires improvement and the positioning of the paper is not clear. Particularly, the method is motivated as an exploration method, and it should be evaluated as such, for example, by comparing to a more representative set of baseline methods. Therefore, I'm recommending rejection, but encourage the authors to improve the work bases on the reviews, and submit to a future conference.